# Retrieval of aerosol optical depth over the Arctic cryosphere during spring and summer using satellite observations

Basudev Swain[1], Marco Vountas[1], Adrien Deroubaix[1,3], Luca Lelli[1,2], Yanick Ziegler[1,4],
Soheila Jafariserajehlou[1,6], Sachin S. Gunthe[5], Andreas Herber[7], Christoph Ritter[8], Harmut Bösch[1], and
John P. Burrows[1]

[1]Institute of Environmental Physics, University of Bremen, Germany
[2]Remote Sensing Technology Institute, German Aerospace Centre (DLR), Wessling, Germany
[3]Max-Planck-Institut für Meteorologie, Hamburg, Germany
[4]now with Karlsruhe Institute of Technology, Institute of Meteorology and Climate Research-Atmospheric Environmental
Research (KIT/IMK-IFU), Garmisch-Partenkirchen, Germany
[5]Indian Institute of Technology Madras, Chennai, India
[6]now with EUMETSAT/Rhea Group, Germany
[7]Alfred-Wegener-Institute, Helmholtz Center for Polar and Marine Research, Bremerhaven, Germany
[8]Alfred Wegener Institute, Helmholtz Centre for Polar and Marine Research, Potsdam, Germany

**Correspondence:** Basudev Swain (basudev@iup.physik.uni-bremen.de)

**Abstract.**

The climate in the Arctic has warmed much faster in the last two to three decades than in the mid-latitudes, i.e., during the Arctic Amplification (AA) period. Radiative forcing in the Arctic is influenced both directly and indirectly by aerosols. However, their observation from the ground or airborne instruments is challenging, and thus measurements are sparse. In this study, total Aerosol Optical Depth (AOD) is determined from top-of-atmosphere reflectance measurements by the Advanced Along-Track Scanning Radiometer (AATSR) aboard ENVISAT over snow and ice in the Arctic using a retrieval called AEROSNOW for the period 2003 to 2011. We use the dual-viewing capability of the AATSR instrument to accurately determine the contribution of aerosol to the reflection at the top of the atmosphere for observations over the bright surfaces of the cryosphere in the Arctic. The AOD is retrieved assuming that the surface reflectance observed by the satellite can be well-parameterized by a bidirectional snow reflectance distribution function (BRDF). The spatial distribution of AODs shows that high values in spring (March, April, May) and lower values in summer (June, July, August) are observed. The AEROSNOW AOD values are consistent with those from colocated AERONET measurements, with no systematic bias found as a function of time. The AEROSNOW AOD in the high Arctic ($\geq 72°N$) was validated by comparison with ground-based measurements at the PEARL, OPAL, Hornsund, and Thule stations. The AEROSNOW AOD value is less than 0.15 on average and the linear regression of AEROSNOW and AERONET total AOD yields a slope of 0.98, a Pearson correlation coefficient of R = 0.86, and a root mean square error (RMSE) = 0.01 for monthly scale, both in spring and summer. The AEROSNOW observation of increased AOD values over the high Arctic cryosphere during spring confirms clearly that Arctic haze events were well captured by this dataset. In addition, the AEROSNOW AOD results provide a novel and unique total AOD data product for the spring and summertime

from 2003 to 2011. These AOD values, retrieved from space-borne observation, provide unique insight into the high Arctic cryospheric region at high spatial resolution and temporal coverage.

## 1 Introduction

The Arctic has experienced a significant increase in near-surface air temperatures over the past three decades: the rate of temperature increase being about four times larger than the global mean (Rantanen et al., 2022). This phenomenon is known as Arctic Amplification (AA). The warming of the Arctic has increased the rate of melting of the Arctic cryosphere, e.g. glaciers, sea ice, and snow-covered surfaces. Processes thought to influence AA include the following: surface albedo feedback (Perovich and Polashenski, 2012), warm air intrusion (Boisvert et al., 2016), and oceanic heat transport (Nummelin et al., 2017), cloud feedback (Kapsch et al., 2013; He et al., 2019; Middlemas et al., 2020), lapse rate feedback (Pithan and Mauritsen, 2014), and effects influencing biological and oceanic particle emission (Park et al., 2015; Campen et al., 2022).

Atmospheric aerosols are a collection of solid or liquid suspended particles that have both natural and anthropogenic sources. It is well known that changes in the scattering and absorption of incoming solar radiation by aerosols have a direct impact on climate change (Bond et al., 2013). Increases in aerosol result in more solar radiation being scattered back into space and the atmosphere and surface are cooled. On the other hand, aerosols that absorb solar radiation warm the atmosphere and surface. Aerosols also act as both cloud condensation nuclei (CCN) and ice nuclei particles (INP), affecting the microphysical and radiative properties of clouds. In this way, aerosols also indirectly affect climate change (Twomey, 1977; Kaufman and Fraser, 1997; Hartmann et al., 2020). However, neither the contribution of aerosols to AA nor the effect of declining regional snow and ice on aerosol during the AA period is well understood (Im et al., 2021).

In this study, we use the Aerosol Optical Depth (AOD) as an optical measure of aerosol. It is valuable in the analysis of the impact of aerosols on the Arctic climate and vice versa. The retrieval of AOD is complicated by the seasonal changes of solar geometry, surface albedo, and meteorology (Mei et al., 2013, 2020a; Stapf et al., 2020). The AOD is defined as the columnar integration of the aerosol extinction coefficient (the sum of the absorption and scattering coefficient).

The Arctic is vast and the ground-based measurements of AOD are inevitably sparse. This has limited our understanding of the direct and indirect impact of aerosols on AA, and vice versa. Recently there have been some campaigns, which have investigated different processes of relevance to aerosol sources and sinks in the Arctic e.g. MOSAiC campaign (Mech et al., 2022), ACLOUD/PASCAL (Wendisch et al., 2023), PAMARCMIPs (Hoffmann et al., 2012; Nakoudi et al., 2018; Ohata et al., 2021). In addition, there are other site-based long-term aerosol measurement studies (Herber et al., 2002; Tomasi et al., 2007; Moschos et al., 2022; Schmale et al., 2022). However, these provide an inadequate spatiotemporal representation of the Arctic region (Sand et al., 2017). The sparseness of AOD may explain, at least in part, the variations of AOD simulations from different climate models (Sand et al., 2017). Without a doubt, the lack of AOD measurements in the Arctic limits our knowledge about radiative forcing and Arctic warming in global and regional climate models (Goosse et al., 2018).

Further, AOD has been retrieved from the measurements of reflectance at the Top-Of-Atmosphere (TOA), made by passive satellite remote sensing instruments over the Arctic, but almost exclusively over snow- and ice-free areas i.e. land and ocean.

A few recent studies have used such AOD products (e.g., (Glantz et al., 2014; Wu et al., 2016; Sand et al., 2017; Xian et al., 2021)) over open ocean and snow- and ice-free surfaces, and make a valuable contribution to closing the data gap mentioned above. However, these AOD products are not suitable over the cryosphere due to an inadequate parameterization of the surface
reflectance over residual snow- and ice-covered areas (Mei et al., 2020a) and Arctic cloud cover (Jafariserajehlou et al., 2019). In addition, typical illumination conditions, i.e. large solar zenith angles make the AOD retrieval used in the Arctic more challenging and lead potentially to a significant overestimation of AOD values (Mei et al., 2013).

Several dedicated algorithms for passive satellite remote sensing over snow and ice have been developed. (Istomina et al., 2009, 2011) and later (Mei et al., 2013, 2020b, a) have provided valuable pioneering research. However, these attempts have
been mostly confined to the island of Spitsbergen in the Svalbard archipelago in northern Norway. Thus far, there have been no attempts to apply these algorithms systematically in the Arctic cryosphere to address the data gap, identified above. Studies using active satellite remote sensing such as Sand et al. (2017) and Xian et al. (2021) are valuable, but the observational data are limited over the Arctic cryosphere.

Recently, Toth et al. (2018) and Xian et al. (2021) reported that the active satellite sensor, the Cloud-Aerosol Lidar with
Orthogonal Polarization (CALIOP/CALIPSO) (Winker et al., 2004) has a significant fraction of aerosol profile data comprising retrieval fill values (-9999s, or RFVs) and thus rejected. This is due in part to the lidar's minimum detection limits of its measurements of the scattered back to space signal. Indeed, in some areas in the Arctic, over 80% of CALIOP profiles consist solely of RFVs.

Consequently, it is potentially of interest to investigate the retrieval of AOD using other techniques, the objective of this
study is to retrieve the AOD over snow- and ice-covered regions of the Arctic by using passive remote sensing from space. We have extended the retrieval of AOD over polar regions for which past efforts, using the same AATSR source data, did not cover the high latitudes (Popp et al., 2016). To achieve this goal, we retrieve the total AOD using an approach first described by Istomina et al. (2009), which we have further developed and named AEROSNOW. We assessed the quality of AEROSNOW by using Aerosol Robotic Network (AERONET) measurements to validate the retrieved AOD. After retrieval and validation,
we discuss the distribution of AOD over Arctic snow and ice in spring and summer, where ground-based and other space-borne observational data on AOD in the high Arctic cryosphere are limited. The purpose of examining these distributions is to gain further confidence in this new data set and to investigate the distributions with respect to our expectations. We examine whether the AOD retrieved by AEROSNOW is able to capture the increased pan-Arctic distribution of AOD in spring compared to summer (Willis et al., 2018), which would be a clear confirmation of whether or not Arctic haze events are well captured by
this dataset.

We have generated the AEROSNOW AOD data set for the period from 2003 to 2011 (9 years). The algorithm uses the measurements, made by the Advanced Along-Track Scanning Radiometer (AATSR) over the Arctic. A short description of the AATSR data and the corresponding retrieval is given in section 2 and the development of the AEROSNOW algorithm is described in section 3.1. To determine the quality of the AEROSNOW data sets, we compare them with accurate ground-based
AERONET measurements in the high Arctic. The description of the AERONET dataset is found in section 2.2. The AERONET

and AEROSNOW values are then compared at selected high latitude AERONET stations over snow and ice-covered surfaces in section 4. Finally, we draw conclusions in section 5.

## 2   Data Sets and Algorithms

To investigate the distribution and variability of Arctic aerosols over snow and ice, we have employed passive remote sensing during spring (March, April, April, May, (MAM)) and summer (June, July, August, (JJA)), which is the time when the Arctic is illuminated by solar radiation.

Prior to the use of an AOD retrieval algorithm suitable for the Arctic, we necessitate a precise, adapted for the Arctic cloud masking scheme. After identifying cloud-free scenes, we applied the AOD retrieval algorithm. To create such an AOD retrieval framework suitable for the Arctic, we integrate two pioneering approaches, which we briefly summarise in the following sections. For cloud masking we utilized the approach of Jafariserajehlou et al. (2019) (section 2.3.1) and for AOD retrieval we used in Istomina et al. (2009) (section. 2.3.2. The integrated framework and subsequent quality flagging (QF) scheme is described in section. 3.1.1, which we have entitled as AEROSNOW.

The AEROSNOW algorithm is applied to the dual view Level 1B data product reflectance at the top of the atmosphere made by AATSR (Llewellyn-Jones et al., 2001). We have validated the AEROSNOW retrieved AOD by comparing it with the AOD measured by the ground-based sun-photometer measurements, AERONET (Holben et al., 1998). We introduce AATSR and AERONET data in Section 2.1 and Section 2.2 respectively.

### 2.1   Space-borne observation: AATSR Instrument

The AATSR flew, as part of the payload of the European Space Agency's (ESA) ENVISAT, which was launched on 28.02.2002 and failed on 08.04.2012. ENVISAT flew in a sun-synchronous orbit with an equatorial crossing local time of 10 o'clock. AATSR made measurements from May 2002 to April 2012. The spatial resolution of the AATSR observation data was 1 km at nadir. The swath width of AATSR was 512 km. AATSR had a dual viewing capability with a forward viewing angle of 55°. It made simultaneous measurements of the upwelling reflectances at wavelengths, from the visible to thermal infrared (0.55, 0.66, 0.87, 1.6, 3.7, 11, and 12 $\mu$m).

### 2.2   Ground-based measurements: AERONET Level 2 Aerosol Product

The AErosol RObotic NETwork (AERONET) is a federated network of ground-based global sun photometers measuring solar and sky irradiance at various wavelengths from the near ultraviolet to the near-infrared with high accuracy (Holben et al., 2001; Giles et al., 2019). AERONET sun photometers record AOD values every 15 minutes in typically seven spectral channels (nominally 340, 380, 440, 500, 670, 870, and 1020 nm) (Holben et al., 2001). The quality-assured AERONET version 3 level 2 data are used in this study (accessed at http://aeronet.gsfc.nasa.gov).

The AOD from AERONET stations in the high Arctic were used to assess the data quality of AOD estimated using AEROS-NOW. The name and locations of the AERONET stations selected are PEARL (80.054N, 86.417W), OPAL (79.990°N,

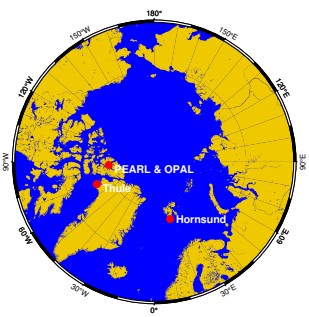

**Figure 1.** Location of PEARL, OPAL, Hornsund, and Thule AERONET measurement stations considered in this study.

85.939°W), Hornsund (77.001°N, 15.540°E) and Thule (76.516°N, 68.769°W) and shown in Fig. 1. Two sites are located over the Canadian archipelago (CA), which typically has aerosol of natural origin (Breider et al., 2017) and one station Hornsund on Spitsbergen, which is known to be affected by polluted air masses transported from lower latitudes.

In addition, The use of data sources other than AERONET for validation would have been very helpful. Unfortunately, data from valuable campaigns and expeditions such as POLAR-AOD (Mazzola et al., 2011), MOSAiC Expedition and MOSAiC-ACA (Mech et al., 2022), and AFLUX/PASCAL - Arctic (Mech et al., 2022) were only available after 2011. For this reason, we have focused on validation with ground-based AERONET measurements.

## 2.3    The Heritage of the Cloud Masking and AOD retrieval algorithms

The heritage of the algorithms used in this study, such as the cloud identification algorithm (Jafariserajehlou et al., 2019), and the aerosol retrieval algorithm (Istomina et al., 2009) comes from earlier studies made at the Institute of Environmental Physics of the University of Bremen. The algorithms are described in the following sections 2.3.1 and 2.3.2, respectively.

### 2.3.1    Cloud detection Algorithm (ASCIA)

The cloud detection algorithm, used in AEROSNOW, is discussed in Jafariserajehlou et al. (2019). The Advanced Along-Track
Scanning Radiometer (AATSR) - Sea and Land Surface Temperature Radiometer (SLSTR) cloud identification algorithm (ASCIA) was developed to address the requirements of Arctic cloud identification. ASCIA is explained in more detail in Jafariserajehlou et al. (2019). Briefly, the ASCIA combines two approaches to detect the presence of clouds: i) a Pearson Correlation Coefficient ($PCC$) analysis of the reflectance at the top of the atmosphere (TOA) at $1.6\mu m$ wavelength, and ii) the use of the reflectance at TOA for a wavelength of $3.7\mu m$. The $PCC$ analysis separates the reflectance of the surface from
cloud reflectance at the TOA. The surface reflectance is stable while the cloud reflectance at the TOA is highly variable over a short period of time. The $PCC$ for two satellite scenes, such as $x$ and $y$, is a function of their covariance, normalized by their standard deviations (Rodgers and Nicewander, 1988; Benesty et al., 2009). It can be interpreted as a measure of how well these two satellite scenes are correlated (Lyapustin et al., 2008).

$$PCC = \frac{COV(x,y)}{\sigma_x \sigma_y}, \quad COV(x,y) = \frac{1}{N} \sum_{i=1}^{N} (x_i - \bar{x})(y_i - \bar{y}) \tag{1}$$

Where, $COV(x,y)$ is the covariance of variables $x$ and $y$ and $\sigma$ is the standard deviation of each variable.

$$\sigma_x = \sqrt{\frac{1}{N} \sum_{i=1}^{N} (x_i - \bar{x})^2}, \quad \sigma_y = \sqrt{\frac{1}{N} \sum_{j=1}^{N} (y_j - \bar{x})^2}. \tag{2}$$

Where the mean values of $x$ and $y$ are $\bar{x}$ and $\bar{y}$, respectively. The $PCC$ has values between -1 and +1 (Rodgers and Nicewander, 1988). The correlation or anti-correlation between the two variables is strongest when the absolute value is closer to 1 or -1, respectively. Consequently, the $PCC$ values are derived using sets of reflectances at TOA of the same area at different times,

which give an indication of whether the scene is cloudy or cloud-free (Lyapustin et al., 2008). The $PCC$ is calculated individually for areas of 25 km x 25 km. Based on statistical analysis, Jafariserajehlou et al. (2019) found a threshold of $PCC \leq 0.6$ as a reliable value for the detection of mid-latitude clouds. This value is also in good agreement with a similar analysis of Lyapustin et al. (2008) using one year of MODIS data for 156 global AERONET stations, where they derived cut-off values of 0.65 to 0.68. In the Arctic, and especially over the cryosphere, surfaces are often found that have low visible or thermal contrast

compared to mid-latitudes. Thus, a specific analysis in the Arctic was performed by Jafariserajehlou et al. (2019) to account for reduced structural patterns due to frequent snow/ice cover. They determined a threshold of $PCC \geq 0.4$ for cloud-free scenes in the Arctic. Having selected cloud-free ground scenes using the $PCC$ on the 25km x 25 km scale Jafariserajehlou et al. (2019) introduced an additional criterion using the reflectance at $3.7\mu m$. They used this channel because the single scattering albedo at 3.7 $\mu$m, compared to that in the VIS and NIR wavelength ranges, is more sensitive to the absorption of liquid water and ice,

or snow (Platnick and Fontenla, 2008).

In summary, a $PCC$ analysis at 25km × 25km scale is performed to identify cloudy and cloud-free scenes that are assumed to have low and high $PCC$ values, respectively. The result of this step are binary flags at 25km × 25km scale. This serves as input to the second step of the algorithm, where the reflectance at TOA for $3.7\mu m$ is used to create binary flags at 1 km × 1 km (AATSR spatial resolution of the instrument at nadir) to identify clouds. The combination of these two constraints is necessary

because neither the $PCC$ analysis nor the reflectance of the $3.7\mu m$ channel alone is sufficient for accurate cloud identification in the Arctic (Jafariserajehlou et al., 2019).

At the $3.7\mu m$ channel, the reflectance of the snow/ice surface (0.005–0.025) has interference with those of ice clouds (0.01–0.3). To avoid the uncertainty arising from this interference, Jafariserajehlou et al. (2019) optimized ASCIA by using thresholds as follows:

i) An area of 25km × 25km with a $PCC \geq 0.4$ is then further analyzed on pixel level (1 km × 1 km) by using the reflectance at TOA at $3.7\mu m$. A scene is considered as cloud-free if the reflectance at TOA at $3.7\mu m$ is larger than 0.04, following Allen Jr et al. (1990).

ii) If an area of 25km × 25km has a value of $PCC < 0.4$ and the reflectance at TOA at $3.7\mu m$ on pixel level is larger than 0.015 the pixel is considered to be cloudy. This threshold is equal to or lower than the lowest observed reflectance of ice clouds at $3.7\mu m$ (Allen Jr et al., 1990).

For more information, the reader is referred to the schematic flow chart of the cloud identification algorithm shown in Fig. 4 of the article by Jafariserajehlou et al. (2019).

In the ASCIA study, the areas identified as cloud-free are assumed to be unchanged for the sampling period of $\pm 30$ min, while cloudy or partly cloudy scenes exhibit much greater spatial and temporal variability (see (Jafariserajehlou et al., 2019)). The cloud fraction retrieved by ASCIA has been validated with SYNOP (WMO, 1995) ground-based cloud fraction measurements, made during the period of AATSR (2003 to 2011) and SLSTR (the year 2018). The World Meteorological Organization (WMO) established SYNOP stations for weather information around the world. The locations of the SYNOP stations used for the ASCIA validation within the Arctic circle are shown in Fig. 3 of Jafariserajehlou et al. (2019). The okta scale is used to report SYNOP cloud fraction. The discrete okta values range from 0 (completely clear sky) to 8 (completely obscured by clouds). The usual assumption is that 1 okta equals 12.5% of cloud coverage. The ASCIA study follows Boers et al. (2010) suggesting a larger range of 18.75% for 1 okta. The use of the okta measurements in ASCIA required the estimation of the error or uncertainty in the measurements. In Boers et al. (2010); Werkmeister et al. (2015), the SYNOP cloudiness okta estimation has errors of $\pm 1$ okta for values of okta between 1 and 7, and a value $\pm 2$ okta for 0 or 8 okta.

The ASCIA cloud detection algorithm achieved promising agreement of more than 95% and 83% within $\pm 2$ and $\pm 1$ okta when compared with ground-based synoptic surface observations (SYNOP) (WMO, 1995) over the Arctic. In general, ASCIA shows a better performance in detecting clouds over Arctic ground scenes than other algorithms applied to AATSR measurements (Jafariserajehlou et al., 2019).

### 2.3.2 AOD retrieval Algorithm

The approach used in our retrieval algorithm was first discussed in Istomina et al. (2009). The reflectance at the top of the atmosphere (TOA), as shown by Chandrasekhar (1950) and Kaufman et. al., (1997), is given by :

$$\rho_{TOA}(\lambda,\mu_0,\mu,\phi) = \rho_{atm}(\lambda,\mu_0,\mu,\phi) + \frac{A_{sfc}(\lambda)T_1(\lambda,\mu_0)T_2(\lambda,\mu)}{1 - A_{sfc}(\lambda)s(\lambda)} \tag{3}$$

Where $\mu = \cos\theta$ and $\mu_0 = \cos\theta_0$, $\theta_0$ and $\theta$ are solar zenith angle and zenith angle of satellite respectively, $\phi$ is the relative azimuth angle, $\lambda$ is the wavelength, $\rho_{TOA}(\lambda,\mu_0,\mu,\phi)$ is the satellite measured to the top of the atmospheric reflectance, $\rho_{atm}(\lambda,\mu_0,\mu,\phi)$ is the contribution of atmospheric reflectance to top of atmospheric reflectance, $A_{sfc}(\lambda)$ is the surface spectral albedo, $T_1(\lambda,\mu_0)$ is the downward transmission of light, $T_2(\lambda,\mu)$ is the surface to TOA transmission of light, and $s(\lambda)$ is the atmospheric hemispherical albedo.

The parameters in Eq. 3 have no apriori values. Thus the separation of the various effects contributing to TOA reflectance is the mathematical problem to be solved. This was achieved by using the BRDF of snow reflectance, the knowledge of the two viewing angles.

As suggested by Flowerdew and Haigh (1995), the surface reflectance can be approximated by two terms: one term describing the variation in wavelength and another describing the variation in geometry. To a first approximation, the ratio between the surface reflectances at two viewing angles depends only on the wavelength. This principle has already been successfully applied in the AATSR-DV algorithm (Veefkind et al., 1998; Curier et al., 2009). Furthermore, (Vermote et al., 1997) determined the ratio between the estimated BRDF and the actual surface BRDF for the atmospheric correction and mentioned that this ratio is only affected by the shape of the BRDF and not by its size. This means that the effect of the magnitude of the surface scatter can be eliminated by using the ratio of AATSR dual-view observations measured in the forward and nadir views (Veefkind et al., 1998; Istomina et al., 2009, 2011).

By assuming that a Lambertian surface, $A_{sfc}(\lambda)$ is equal to the surface reflectance $\rho_{sfc}(\lambda)$ using Eq. 3 we obtain

$$\frac{\rho^f_{sfc}(\lambda,\mu_0,\mu,\phi)}{\rho^n_{sfc}(\lambda,\mu_0,\mu,\phi)} = \frac{\rho^f_{TOA}(\lambda,\mu_0,\mu,\phi) - \rho^f_{atm}(\lambda,\mu_0,\mu,\phi)}{\rho^n_{TOA}(\lambda,\mu_0,\mu,\phi) - \rho^n_{atm}(\lambda,\mu_0,\mu,\phi)} \cdot \frac{T^n(\lambda,\mu)}{T^f(\lambda,\mu)} \tag{4}$$

where $\rho_{sfc}(\lambda,\mu_0,\mu,\phi)$ is the reflectance from the surface, $T(\lambda,\mu) = T_1(\lambda,\mu_0)T_2(\lambda,\mu)$ is the total atmospheric transmittance from the surface to a satellite sensor, and $f$ and $n$ indicate AATSR forward and nadir observation angles, respectively.

BRDF ratio can be described by the left term of Eq. 4. For the estimation of this ratio, the two-parameter snow BRDF model (Kokhanovsky and Breon, 2012) has been used to parameterize the snow spectral reflection function $\rho(\mu,\mu_0,\phi)$:

$$\rho(\mu,\mu_0,\phi) = \rho_0(\mu,\mu_0,\phi) \times e^{[-\psi K_0(\mu)K_0(\mu_0)/\rho_0(\mu,\mu_0,\phi)]} \tag{5}$$

$$\rho_0(\mu,\mu_0,\phi) = \frac{a + b(\mu+\mu_0) + c\mu\mu_0 + p(\theta)}{4(\mu+\mu_0)} \tag{6}$$

where

$$K_0(\mu) = \frac{3}{7}(1+2\mu), K_0(\mu_0) = \frac{3}{7}(1+2\mu_0) \tag{7}$$

$$a = 1.247, b = 1.186, c = 5.157 \tag{8}$$

$$p(\theta) = 11.1\exp(-0.087\theta) + 1.1\exp(-0.014\theta) \tag{9}$$

$$\cos\theta = -\mu\mu_0 + ss_0\cos\phi \tag{10}$$

$$s = \sqrt{1 - \mu^2}, s_0 = \sqrt{1 - {\mu_0}^2} \qquad (11)$$

$$\psi = \sqrt{\gamma L} \qquad (12)$$

Where $\gamma = 4\pi \times (\chi + M) \times \lambda^{-1}$, $\chi$ is the imaginary part of the refractive index of ice, $\lambda$ is the wavelength, $L$ is related to snow grain size, and $M$ is related to the estimation of the absorption of light by pollutants (Kokhanovsky and Breon, 2012).

Some corrections are needed in order to take the snow structure into account as only pure snow-covered area (100% snow cover) is used for the retrieval and in the real AATSR measurements. The way to take this effect into account is proposed by Mei et al. (2012). The primary drive of the idea is that by using a correction term by the snow cover fraction (SCF), the "real snow" BRDF can be better estimated from the original BRDF model (Kokhanovsky et al., 2005).

    The Normalized Difference Snow Index (NDSI) is defined as the following:

$$NDSI = \frac{\rho_{0.55} - \rho_{1.6}}{\rho_{0.55} + \rho_{1.6}} \qquad (13)$$

    The NDSI is an index that refers to the presence of snow in a pixel and is a more accurate description of snow detection compared to fractional snow cover. Snow typically has a very high reflectance in the visible spectrum (VIS) and a very low reflectance in the shortwave infrared (SWIR). Snow coverage is determined by the NDSI ratio of the difference between the VIS and SWIR reflectance, and is defined in Eq. 13.

The SCF was then estimated using the empirical equation (Salomonson and Appel, 2004):

$$SCF = 1.21 \cdot NDSI + 0.06 \qquad (14)$$

    Then, the snow BRDF ratio can be rewritten as follows:

$$\frac{\rho_{sfc}^f(\lambda, \mu_0, \mu, \phi)}{\rho_{sfc}^n(\lambda, \mu_0, \mu, \phi)} = \frac{\rho_{sfc,sim}^f(\lambda, \mu_0, \mu, \phi) \cdot SCF^f}{\rho_{sfc,sim}^n(\lambda, \mu_0, \mu, \phi) \cdot SCF^n} \qquad (15)$$

    Where, $\rho_{sfc,sim}(\lambda, \mu_0, \mu, \phi)$ is the simulated top of the atmospheric reflectance for a given AOD. Using this equation, BRDF

is obtained with an uncertainty of 15% ((Kokhanovsky and Breon, 2012).

    The BRDF model is analytical and has been compared with a set of multispectral and multidirectional measurements from the POLDER-3 (Polarization and Directionality of the Earth's Reflectances) instrument aboard the PARASOL (Polarization and Anisotropy of Reflectances for Atmospheric Sciences coupled with Observations from a Lidar) satellite. Istomina et al. (2009) fixed the free parameters for the entire time series of AATSR, which involved a fixed snow grain size and snow impurity

assumptions. The BRDF model reproduces the directional variations in the measured reflectance with a root mean square (RMS) error that is typically 0.005 in the visible wavelength range (Kokhanovsky and Breon, 2012), but The accuracy of the BRDF ratio is sufficient to represent the aerosol effect. So, Istomina et al. (2009) assumes that they can accurately estimate the BRDF ratio using Eq. 3 to Eq. 12.

     Another objective is how to estimate reflectance from the atmosphere $\rho_{atm}(\lambda, \mu_0, \mu, \phi)$ in Eq. 4. This can be achieved

by using reflectance over the ocean as a first estimate, as assumed by Kokhanovsky and Schreier (2009). The AATSR TOA reflectance at 555 nm over the ocean around Greenland is 0.084 reported by Kokhanovsky and Schreier (2009). By using SCIATRAN radiative transfer model the water-leaving radiance was calculated at 555 nm is 0.014. Hence, 0.07 is estimated as the atmospheric contribution to this specific case. In the absence of sources except for sea spray aerosol, which is mainly driven by high wind speeds and in the surf zone (Leeuw, 1999), a smooth transition of AOD across the coastal zone is assumed.

This value is also used for atmospheric reflection over the coastal region of Greenland. In Section 3 of (Mei et al., 2013), the validity of this assumption used for the cases studied here is evaluated.

     Similar to Kokhanovsky and Schreier (2009), Istomina et al. (2009) assumed the variability of $\rho_{atm}(\lambda, \mu_0, \mu, \phi)$ is small for different observational and solar illumination conditions in the Arctic. Using the Arctic haze phase function shown in Fig. 2 (Istomina et al., 2009), the simulations were made. Two types of aerosol phase functions were used in these simulations: Arctic

haze and background aerosol. In Fig. 2, the phase functions for these two types are shown.

     Over the study area, for solar zenith angles between 45° and 65°, the simulations by (Mei et al., 2013) showed that the $\rho_{atm}(555nm, \mu_0, \mu, \phi)$ ranges from 0.058 to 0.079 (AOD=0.2). (Mei et al., 2013) assumed that cloud-free scenes over the ocean have the smallest $\rho_{atm}(555nm, \mu_0, \mu, \phi)$ and the $\rho_{sfc}(\lambda, \mu_0, \mu, \phi)$ is negligible then $\rho_{atm}(555nm, \mu_0, \mu, \phi)$ is equal to $\rho_{TOA}(555nm, \mu_0, \mu, \phi)$.

As a result the smallest values of $\rho_{TOA}(555nm, \mu_0, \mu, \phi)$ in the above range over the ocean were used to determine $\rho_{atm}(\lambda, \mu_0, \mu, \phi)$. Further, the approximated values given by Kokhanovsky and Schreier (2009) were used, if no pixel in the scene under consideration could be found with$\rho_{toa}(\lambda, \mu_0, \mu, \phi)$ in this range. Kokhanovsky et al. (2005) presented the relationship between AOD and transmissivity. An air mass correction is needed in the BRDF calculation at the high solar zenith angle in the Arctic region.

Similar to Kasten and Young (1989), the air mass factor m(z) was calculated as follows:

$$m(z) = [\cos(z) + A(B - C)]^{-1} \tag{16}$$

     where $A = 0.050572, B = 6.07995, C = 1.6364$

     As is well known the Ångström equation or relationship (Ångström, 1929) enables AOD to extrapolate the AOD measured at a given $\lambda$ to another $\lambda$ :

$\tau = \beta\lambda^{-\alpha}$                                                (17)

In Istomina et al. (2009), aerosol optical depth is derived using the equations from Eq. 4 to Eq. 17 and an iterative procedure involving observations at wavelength 555nm to minimize the cost function, $S(\alpha, \beta, L, M)$, given by:

$$S(\alpha, \beta, L, M) = \|\frac{\rho_{sfc}^{f}(\lambda, \mu_0, \mu, \phi, L, M)_{555nm}}{\rho_{sfc}^{n}(\lambda, \mu_0, \mu, \phi, L, M)_{555nm}} - \frac{\rho^{f}(\mu_0, \mu, \phi, \alpha, \beta, L, M)_{555nm}}{\rho^{n}(\mu_0, \mu, \phi, \alpha, \beta, L, M)_{555nm}}\|^2 \tag{18}$$

The unknown parameters, such as $\alpha, \beta, L, M$, are retrieved from Eq. 18 for wavelength 555nm. This is achieved by applying a root-finding algorithm (Brent, 1971).

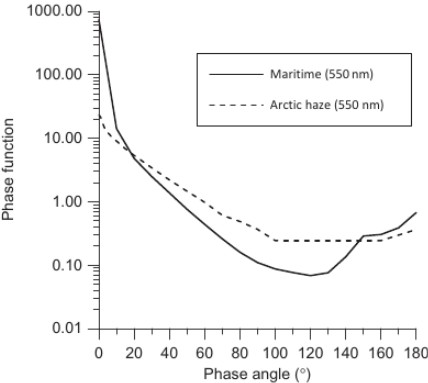

**Figure 2.** Phase function from ground-based measurement at $0.55\mu m$ during the Arctic haze event on 23 March 2003, at Spitsbergen (78.923°N, 11.923°E) (Istomina et al. 2009). For comparison, the phase function for maritime aerosol is also shown.

The aerosol properties used in the model, such as the single scattering albedo (SSA ($\lambda$)), the real part, and the imaginary part of the refractive index for the coarse and accumulation modes of the water-soluble, oceanic, dust, and soot aerosol components are given in Table. 1 adopted from Istomina et al. (2011). Further, in Istomina et al. (2009), they used the phase function for 555 nm, which they assumed is the same as that at 555 nm and was measured by the Alfred Wegener Institute for Polar and Marine Research during the Arctic haze event on 23rd March 2000 at Spitsbergen, Ny Ålesund, Svalbard, 78.923°N, 11.923° E. Subsequently, the look-up table was calculated using the SCIATRAN radiative transfer package (Rozanov et al., 2014; Mei et al., 2023a).

A combination of seven AATSR channels was used to distinguish the spectral response of a clear, snow-covered landscape from that of clouds, land, and sea using the visible (Eq. 20, Eq. 21) and near-infrared (Eq. 19) nadir reflectances of the top of the atmosphere. While the visible (VIS) and near-infrared (NIR) criteria. (Eq. 19, Eq. 20, 21) select scenes whose spectral behavior is similar to the snow spectrum (Istomina et al., 2009).

$$\frac{\rho_{atm}(0.87\mu m, \mu_0, \mu, \phi) - \rho_{atm}(1.6\mu m, \mu_0, \mu, \phi)}{\rho_{atm}(0.87\mu m, \mu_0, \mu, \phi)} > 80\% \tag{19}$$

| | Refractive index | | Single Scattering Albedo | |
|---|---|---|---|---|
| Aerosol component | $n$ | $i$ | Coarse mode | Accumulation mode |
| Water soluble | 1.530 | 6.00E-03 | 0.75 | 0.92 |
| Oceanic | 1.381 | 4.26E-09 | 1.00 | 1.00 |
| Dust | 1.530 | 8.00E-03 | 0.71 | 0.89 |
| Soot | 1.750 | 4.40E-01 | 0.55 | 0.50 |

**Table 1.** Single scattering albedo (SSA), real '$n$' and imaginary '$i$' part of the refractive index for coarse and accumulation mode of water-soluble, oceanic, dust and soot aerosol components at 555 nm (Istomina et al., 2009)

$$\frac{\rho_{TOA}(0.87\mu m) - \rho_{TOA}(0.66\mu m)}{\rho_{TOA}(0.87\mu m)} < 10\% \tag{20}$$

$$\left| \frac{\rho_{TOA}(0.66\mu m) - \rho_{TOA}(0.55\mu m)}{\rho_{TOA}(0.66\mu m)} \right| < 10\% \tag{21}$$

Where $R_{TOA}$ is the reflection at the top of the atmosphere.

## 3 Methodology

### 3.1 AEROSNOW

The dual-viewing capability of the AATSR instrument was used to retrieve AOD over the pan-Arctic snow and ice region. Before an AOD retrieval algorithm suitable for the Arctic can be used, we need a rigorous Arctic-adopted cloud detection
algorithm. Therefore, in this study, we developed the AEROSNOW framework for pan-Arctic AOD retrieval. The development of AEROSNOW involves the integration of two different algorithms, the first algorithm being the cloud detection algorithm of Jafariserajehlou et al. published in 2019, which is described in section 2.3.1, with the core AOD retrieval algorithm of Istomina et al. published in 2009, as the second algorithm, which is described in section 2.3.2, followed by a quality flagging described in following section 3.1.1.
After formulating the AEROSNOW framework, we applied it systematically for the first time to the pan-Arctic cryosphere region to obtain space-borne observational data with high spatial and temporal coverage. The flowchart describes the important building blocks of AEROSNOW presented in Fig 3.

### 3.1.1 Data quality flagging

Quality flagging of the AOD data product gives information about our assessment of the accuracy of the AOD determined using
AEROSNOW. To compare the AOD of AERONET, measured at 500 nm with those of AEROSNOW, measured at 555 nm, a

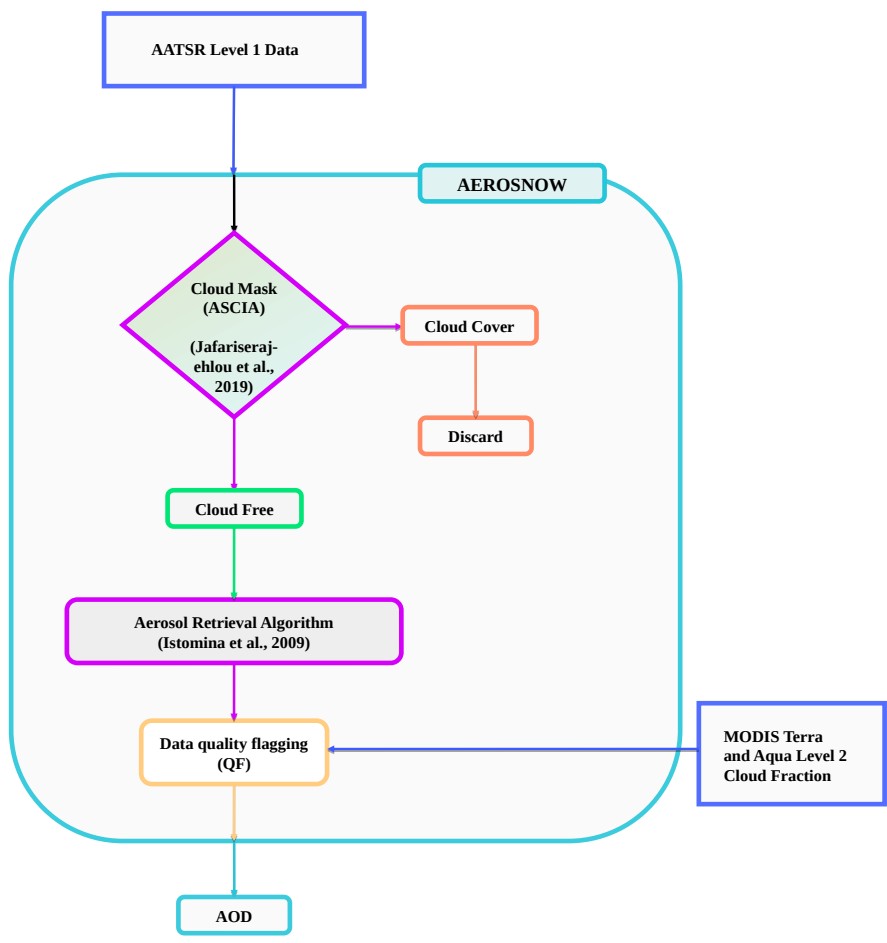

**Figure 3.** Flowchart describing the important building blocks of the AEROSNOW scheme.

conversion is required. The AOD at 500 nm is extrapolated to the AOD at 555 nm by using the Angstrom Exponent determined in the 500 - 870 nm region. We consider the AOD at 555 nm to be the total AOD of the Atmosphere. This was validated by comparisons with AERONET AOD. For these comparisons, the AEROSNOW observations were averaged within a 25-km radius of the AERONET station and within a ±30-minute period of the AERONET measurement. Monthly averages of these

AEROSNOW spatio-temporal co-located data were generated. Information about the collocated daily values derived is shown in the appendix (Fig. A1).

In post-processing AEROSNOW, we selected optimal conditions, i.e. by using a cutoff value for the solar zenith angle ( $\geq 75°$) and filtering out those scenes or pixels that had a Normalized-Difference Snow Index (NDSI) $\leq 0.97$. In this work, we adopted the recommendations in (Istomina (2011), Section 3.3.5) for a solar zenith angle (SZA) of 75 degrees based on

sensitivity analysis using the SCIATRAN radiative transfer model (see also (Mei et al., 2023b)).

A pixel with NDSI > 0.0 is considered to have snow present, while a pixel with NDSI ≤ 0.0 represents a snow-free land surface (Riggs et al., 2017). In this study, NDSI was used in a rigorous post-processing of the datasets to filter out the mixed and snow-free regions. For the stations considered in this study, we found bimodal distributions of AOD, with the frequency of the second mode influenced by clouds and problematic surfaces. We filtered out 40% over problematic surfaces and residual clouds and 100% over central Greenland using rigid post-processing with NDSI. The AEROSNOW retrieval is based on Eq. 4 to Eq. 17, which uses the ratio of the simulated nadir BRDF values for the nadir and forward views of the dual-viewing instrument AATSR. With this strategy, we mitigate the absolute errors in the BRDF but rather rely on the shape of the BRDF as seen from both directions. For our study, a narrow interval of NDSI was required to limit the BRDF-induced error in retrieving the AOD, which is less than 30% using Istomina's approach (Istomina (2011), Section 3.3.3). Since we consider this error to be critical, we additionally introduced the Quality Flagging (QF) approach in our post-processing scheme by adopting independent additional support from MODIS Terra and Aqua cloud fraction (Ackerman et al., 2008) apart from ASCIA cloud detection algorithm, where we weighted the snow cover fraction even higher than the cloud fraction. Further, as per Liu et al. (2022) for the MODIS Terra and Aqua CF products over the Arctic, the discrepancies caused by different sensors and different algorithms are ±2% and ±5% with respect to the International Arctic Atmospheric Observing Systems (IASOA). The exact locations of the IASOA observatories are shown in Fig. 1 of (Uttal et al., 2016).

To ensure quality, we also noted that melt ponds and thin clouds not captured by rigid filtering deteriorate the retrieval quality of AEROSNOW. Recently Xian et al. (2021) introduced a post-processing scheme to remove erroneous AOD outliers. Instead of applying the rather qualitative quality flagging approach as proposed by the former, we applied a more quantitative one. We thoroughly examined the ratio of AEROSNOW to AERONET AOD greater than a factor of 1.6 by calculating a quality flag (QF) parameter [QF = (0.8 × (snow cover fraction) + 0.2 × (1 - cloud fraction)] which penalizes the low levels of snow cover and (and to less extent) residual cloud fraction by using MODIS Terra and Aqua data products from NASA Worldview (https://worldview.earthdata.nasa.gov/) respectively. When using the AEROSNOW data beyond the AERONET stations, we applied this scheme to all data points of AEROSNOW. Empirical tests showed that an appropriate snow cover fraction is weighted higher than an appropriate clear sky fraction (1 - cloud fraction). This is expressed by the individual weighting factors for snow cover fraction (0.8) and clear sky fraction (0.2). We found that a QF threshold value of 0.6 represents a compromise between data yield and data quality. Thus, in the final step, AOD values having a QF value of 0.6 or less were removed.

## 4 Result: Assessment of AEROSNOW AOD

### 4.1 Qualitative Analysis of AEROSNOW

Before we turn to quantitative validation of the AEROSNOW results, we discuss the spatio-temporal distribution of the AEROSNOW data here briefly in qualitative terms. The spatio-temporal frequency of observations over the Arctic from both ground and satellite is greater in summer (JJA) than in spring (MAM). Fig. 4(a) shows the monthly averaged AOD over snow

and Arctic ice for the period 2003-2011, with significant differences in the spatial distribution of AOD. Fig. 4(b) shows the number of pixels used to average AOD per grid cell during 2003-2011 for March through October.

In this study, satellite retrievals are performed only when snow and ice are present (NDSI-threshold values, see section 2). The NDSI was used to constrain the BRDF in terms of snow grain size and impurity (when snow accumulation is fresh) and clouds are absent. Accordingly, the best coverage is obtained over persistently homogeneous areas covered with (fresh) snow and ice. Greenland is an exception to the AOD retrieval. Possible reasons for this could be that the BRDF does not fit well because it does not adequately represent the snow grains and impurities of the Greenland glaciers and snow covered ice sheet and the elevated topography. Further, the clouds over Greenland are typically optically thin and low-hanging (Bennartz et al., 2013) and are likely not all captured by the ASCIA cloud detection algorithm. Smoother patterns are observed in Fig. 4 over the Arctic sea ice compared to snow-covered land.

## 4.2 Statistical evaluation of AODs from AERONET and AEROSNOW

The satellite retrieved and ground based observations are compared. Fig. 5 shows the validation between AEROSNOW re-trieved AOD over PEARL, OPAL, Hornsund and Thule, with AERONET AOD during this study period. Fig. A1 in the appendix depicts daily collocated values and statistics.

Much of the data analysis in this paper involves fitting a straight line to the AODs observed by both AERONET and AEROS-NOW. Because the AOD observations by these two platforms are both subject to measurement errors (Sinyuk et al., 2012; Mei et al., 2013), we have used a fitting procedure known as the reduced major axis (RMA) method, as described by Hirsch and Gilroy (1984); Ayers (2001).

RMA regression takes into account the uncertainties or errors of the two variables, while ordinary least squares (OLS) regression takes into account the uncertainties of one variable. Here, the best linear fit between the two variables is sought, which must be the same in X or Y regardless of the variable, that is, aiming for a symmetric relationship. For this, different methods are possible, for example, by minimizing the perpendicular distance or the triangle, and we choose the RMA, that is, by minimizing the triangle. RMA has been used by other researchers in the analysis of air quality and atmospheric chemistry data see for example Keene et al. (1986); Arimoto et al. (1995); Freijer and Bloemen (2000); Ayers (2001); Wang et al. (2004)

By combining these four stations, the coefficient of correlation (R) is on average 0.86 and the root mean square error (RMSE) is 0.01. Validating each stations separately the R-value is increasing to 0.90, 0.94, 0.81, and 0.87 over PEARL, OPAL, Hornsund, and Thule respectively (Fig. 6). We consider R values of about 0.8 to be sufficient under the Arctic condition that these are exceptionally challenging retrievals.

## 4.3 Comparison of AODs from AERONET and AEROSNOW

In the next step, we analyze the colocated temporal evolution of the AEROSNOW results and compare them with AERONET station data. The time series of retrieved AEROSNOW AOD is shown together with AERONET data in Fig. 7. In general, the time series for AERONET is well reproduced by AEROSNOW.

In general and as expected, we observe that PEARL, OPAL, and Thule (extended Canadian archipelago, henceforth called CA-stations) exhibit similar temporal behavior. PEARL and OPAL AERONET sites are located around 11.5 km apart. OPAL is closer to the coast than PEARL station and are located at altitudes of 5.0 m and 615.0 m, respectively. Hornsund is in this context clearly different. This difference can be explained by using a chemical transport model by separating the total AOD to aerosol components (Breider et al., 2014).

The CA-stations show low average values AOD. This is associated with Arctic background scenarios. The seasonal variability over all these three stations is presented in Fig. 8, which shows that both the AEROSNOW and AERONET results exhibit higher AOD during spring (MAM) and minimum during summer (JJA). A partially high estimation of AEROSNOW is observed over all the selected sites in August, which may be due to uncertainties in surface parameterization and aerosol types in this region.

On average, AEROSNOW appears to capture some haze events during spring and thereby has higher average values than in the summer season.

## 4.4   Spring and Summertime AOD over the Arctic Sea Ice

Similar to the analysis shown in Fig. 7, where we examined the time series of total AOD from AEROSNOW over AERONET stations, we now discuss qualitatively how AOD values change over time across the entire region of Arctic sea ice.

The AEROSNOW AOD results over pan-Arctic sea ice shown in Fig. 9 and Fig. 10, exhibit maximum values in spring 2009 and minimum values in spring 2006. However, comparing the seasonally averaged climatology from 2003 to 2011, the AEROSNOW results indicate higher AOD in the spring, and smaller values in summer as shown in Fig. 9 and Fig. 11, which was expected due to the Arctic haze events (Willis et al., 2018).

     On the other hand, the AEROSNOW retrievals of AOD may be affected by cloud contamination because high levels of
cloud cover are observed over the Arctic in summer with average values around 0.8 (Kato et al., 2006). Although we adopted reasonable cloud masking for the AOD retrievals, we cannot exclude entirely the possibility that residual cloud contamination prevails (Jafariserajehlou et al., 2019). Additionally, AEROSNOW captures the higher values of AOD over north Alaska and Siberia during summer, a region which is often influenced by boreal forest fires during this period (Xian et al., 2021) (Fig. 9). During spring the higher values of AOD (0.1-0.12) are observed near Europe and the Asian continent and smaller values
(0.07-0.08) towards CA.

     Spring values are mostly dominated by long-range transport of anthropogenic aerosols from the lower latitudes of Europe, America, and Asia (Willis et al., 2018). The AEROSNOW estimates of AOD over central Arctic sea ice are reasonable and thus a valuable source to fill the aerosol data deficit over the perennial sea ice region in the high Arctic.

## 5   Conclusions

This is the first time that AOD has been retrieved using satellite data over the entire Arctic snow and ice surface over a nine-year period and validated with ground based AERONET measurement. A satellite-based retrieval of AOD over Arctic snow and ice

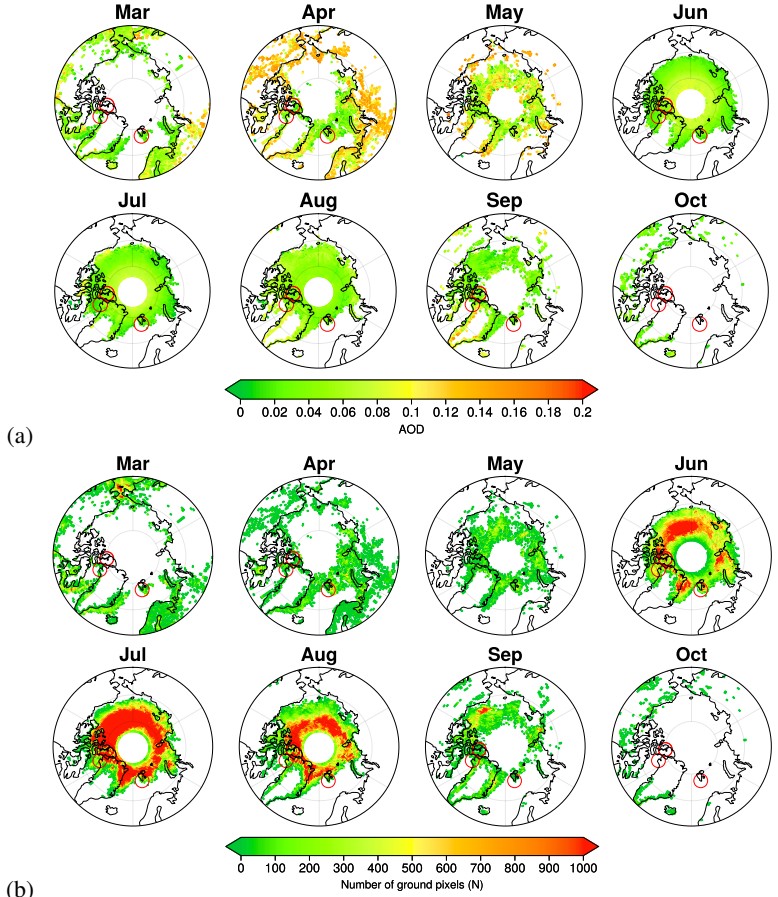

**Figure 4.** (a) Pan-Arctic seasonal view of AEROSNOW retrieved AOD over snow and ice averaged from the year 2003 to 2011 for the months March to October, thus large parts of the period of insolation. Red circles indicate the location of AERONET stations. (b) Pan-Arctic seasonal view of AEROSNOW retrieved the number of ground pixels (N) constituting the monthly average AOD over snow and ice averaged from the year 2003 to 2011 for the months March to October, thus large parts of the period of insolation. Red circles indicate the location of AERONET stations for guidance. The size of the red circles is not identical to the spatial collocation radius of 25 km, which we have used in the validation.

(AEROSNOW, (Istomina et al., 2009, 2011)) was conducted, which has been shown to fill the gap in data availability from standard aerosol products such as e.g. MODIS. The AEROSNOW algorithm uses the dual-viewing capability of the AATSR instrument to minimize retrieval uncertainties (Istomina et al., 2009, 2011; Mei et al., 2013). It showed good agreement with ground-based AERONET observations, with a correlation coefficient R = 0.86 and a low systematic bias.

The high anthropogenic aerosol loading (Arctic haze events) due to long-range transport (Willis et al., 2018) over Arctic snow and ice is captured by the AOD determined by AEROSNOW. Further, the monthly mean spatial maps shown in Fig. 4 confirmed that the haze events are captured well. The time series and seasonality of the AEROSNOW AOD agree well with

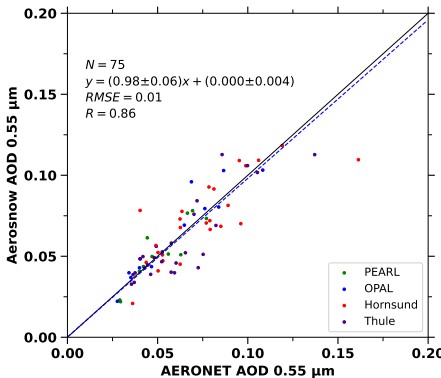

**Figure 5.** Validation of monthly mean AEROSNOW retrieved AOD colocated with monthly mean AERONET observation AOD obtained over PEARL, OPAL, Hornsund, and Thule stations. The linear regression line is shown as the blue dashed line.

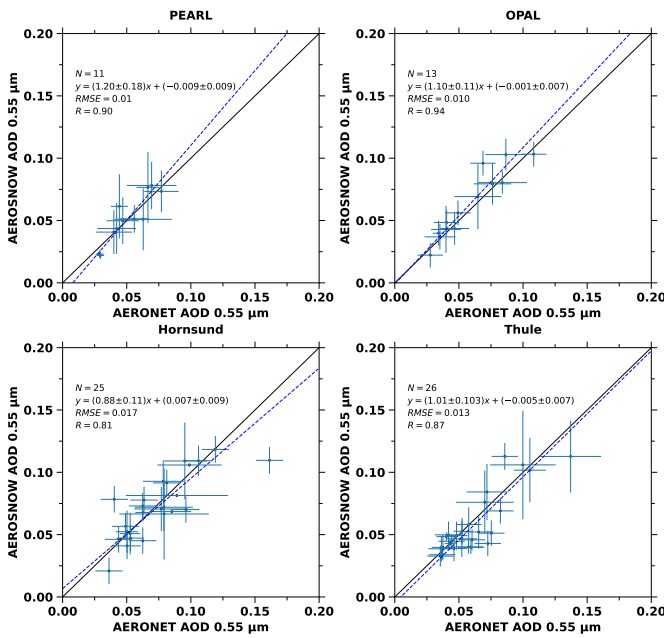

**Figure 6.** Validation of monthly mean AEROSNOW retrieved AOD colocated with monthly mean AERONET observation AOD obtained over PEARL, OPAL, Hornsund, and Thule stations. The linear regression lines are shown as blue dashed lines.

AERONET observations. A good agreement between the AEROSNOW and AERONET AOD is achieved over PEARL, OPAL,
Hornsund, and Thule stations. The AEROSNOW retrieved AOD shows maximum values during spring (MAM) and minimum during summer (JJA), which is also in accordance with AERONET measurements, which shows a clear confirmation that Arctic haze events were well captured by the AEROSNOW retrieved AOD. Further improvement of the AOD retrieval could

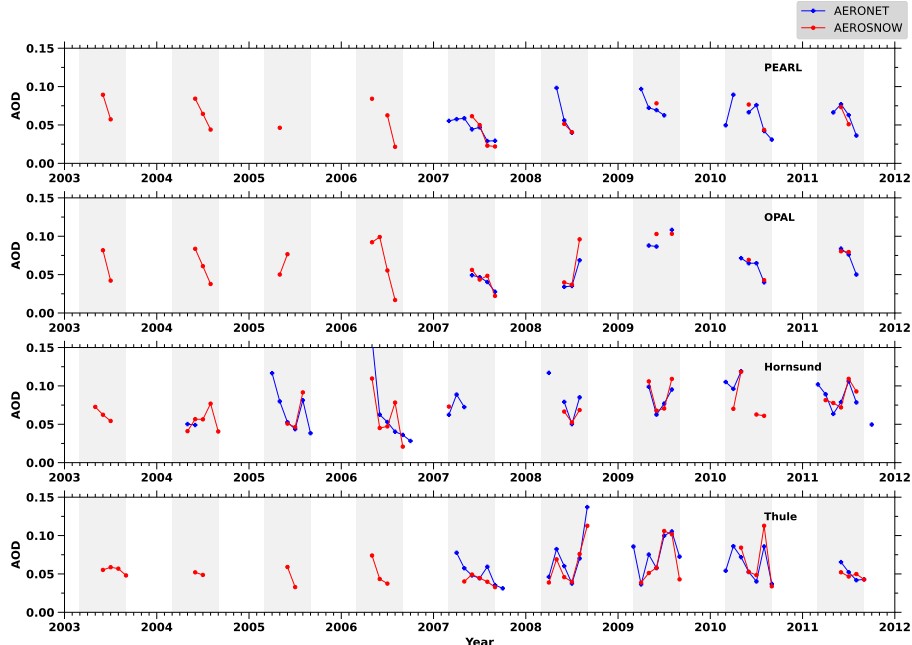

**Figure 7.** Monthly mean time-series of AERONET and AEROSNOW AOD at PEARL, OPAL, Hornsund, and Thule stations. The MAM and JJA periods are highlighted with light grey shades.

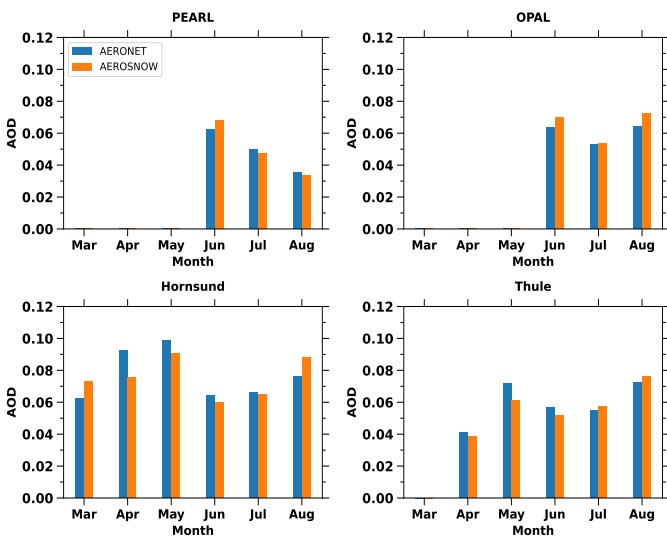

**Figure 8.** Seasonal AOD variation over PEARL, OPAL, Hornsund, and Thule with AEROSNOW and AERONET values average from 2003 to 2011. Blue and orange bar plots show statistics for AERONET and AEROSNOW respectively.

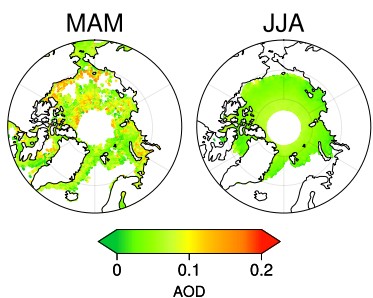

**Figure 9.** Mean climatological MAM and JJA AEROSNOW derived AOD over Arctic Sea Ice averaged from the year 2003 to 2011. The white area shows a lack of data apart from the masked land area.

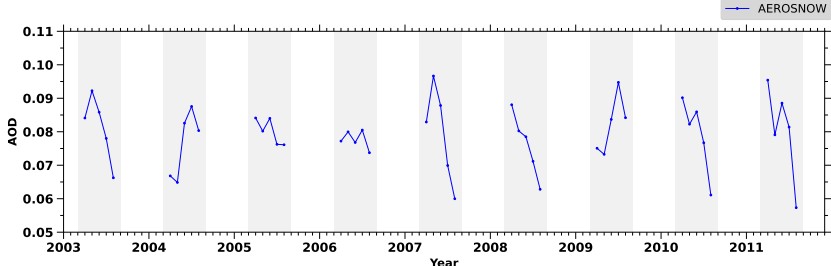

**Figure 10.** Monthly mean time-series of AEROSNOW AOD over Arctic sea-ice region. The MAM, and JJA periods are highlighted with light grey shades.

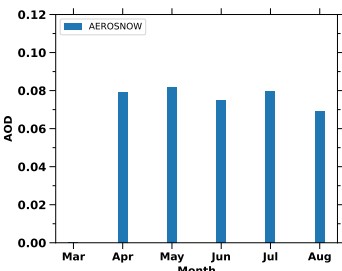

**Figure 11.** Seasonal AOD variation over Arctic sea-ice region values averaged from 2003 to 2011. Blue bars denote the observed monthly mean AODs of AEROSNOW.

be possible in terms of cloud masking, surface reflectivity properties, and the adoption of more appropriate aerosol types to be considered.

The promising AOD results obtained with AEROSNOW indicate that these can be used to evaluate and improve aerosol predictions for various chemical transport models (Willis et al., 2018), especially over the Arctic sea ice in spring and summer for the important period 2003-2011, which is within the period of Arctic amplification.

# 6 Appendix A: Additional Figures

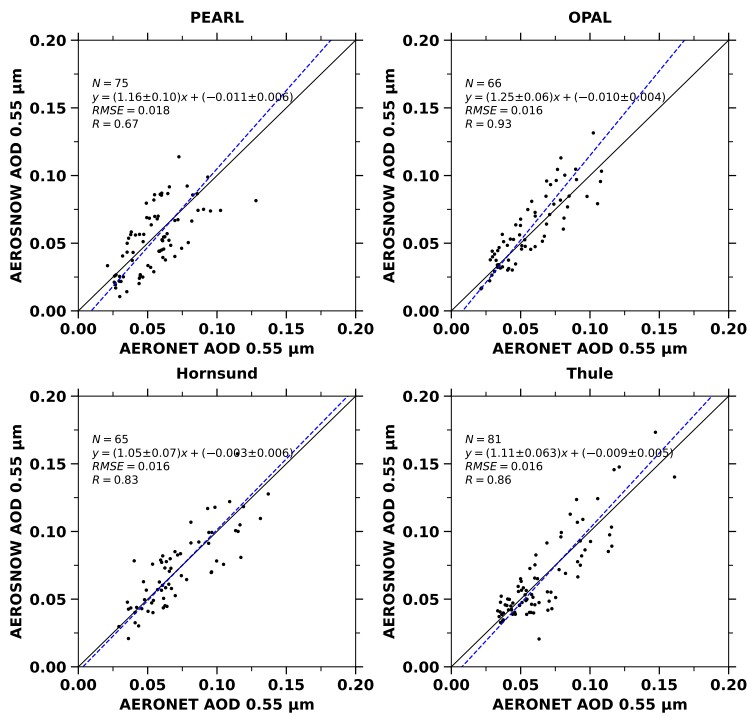

**Figure A1.** Validation of daily AEROSNOW retrieved AOD colocated with daily AERONET observation AOD obtained over PEARL, OPAL, Hornsund, and Thule stations. The linear regression lines are shown as blue dashed lines.

*Code and data availability.* The code and data supporting the conclusions of this manuscript are available upon request.

*Author contributions.* B.S., M.V. conceived the research. B.S. has processed the aerosol data, analyzed all records and wrote the manuscript. S.J. helped in algorithm development. M.V., A.D., L.L, Y.Z, S.J, S.S.G, A.H, C.R, H.B, and J.P.B. helped in shaping this manuscript. Funding acquisition by M.V. and J.P.B. All authors contributed to the interpretation of the results and the final drafting of the paper.

*Competing interests.* The authors declare that they have no conflict of interest.

*Acknowledgements.* We thank ESA for AATSR data set. This work has been funded by the Deutsche Forschungsgemeinschaft (DFG, Ger-
man Research Foundation) within the project "ArctiC Amplification: Climate Relevant Atmospheric and SurfaCe Processes, and Feedback Mechanisms (AC)[3]" as Transregional Collaborative Research Center (TRR) 172, Project-ID 268020496.

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
