# Peer review of "Retrieval of aerosol optical depth over the Arctic cryosphere during spring and summer using satellite observations"

_Atmospheric Measurement Techniques, 2023_

## Author Comment (AC1)

**Previous title:** Spring and summertime aerosol optical depth retrieval over the Arctic cryosphere by using satellite observations

**Revised title:** Retrieval of aerosol optical depth over the Arctic cryosphere during spring and summer using satellite observations: A validation and evaluation

The authors thank the referee for her/his effort, and time taken to review our manuscript. The valuable criticisms and comments have helped us to improve our paper. We hope that we have been able to answer satisfactorily the questions raised and clarify parts of the manuscript which were unclear or ambiguous.

We have changed the title of the manuscript to make it more interesting to AMT's readership.

In the following the referee comments and criticisms, our responses, as authors, and our resultant changes to the manuscript are colored black, blue and red respectively.

**Q1:** The article titled "Spring and summertime aerosol optical depth retrieval over the Arctic cryosphere by using satellite observations" provides an analysis of AEROSNOW-retrieved satellite aerosol optical depth (AOD) statistics and validation using AERONET data over the Arctic region from 2003 to 2011. However, the main objective of the study is not clearly defined. The introduction highlights the importance of AOD retrieval using passive sensors, which suggests an algorithm development focus, which is suitable for the AMT journal. Nonetheless, the majority of the article discusses the distribution of retrieved AOD over the Arctic region and comparisons against AERONET data, with a greater emphasis on understanding the AOD distribution during spring and summer. The content of the article would be better suited for journals with a stronger emphasis on scientific aspects rather than technical aspects, as found in AMT. If the article intends to evaluate the accuracy and uncertainty of the retrieved AEROSNOW AOD, further investigation into uncertainties and comparisons with field campaign data would be necessary.

**Response:** We regret that the description in lines 65 to 69 of the manuscript did not convey to the referee clearly enough our objectives. Our main objective of the study described in this manuscript is to retrieve the aerosol optical depth (AOD) over the pan-Arctic cryosphere using a novel optimized algorithm, AEROSNOW, applied to measurements of the reflectance at the top of the atmosphere, measured by AATSR, made from the low earth orbit sun synchronous satellite Envisat from 2003 to 2011. Previous studies, retrieving AOD using the measurements of passive remote sensing instrumentation from space by others do typically not provide values over the pan-Arctic cryosphere region. So far, earlier versions of the retrieval were used to test the ability to retrieve AOD above Spitsbergen/Svalbard. In this manuscript, we describe an improved algorithm originally developed in house at the Institute of Environmental Physics at the University of Bremen and published by Istomina et al. (2009, 2011). As we have retrieved AOD over the cryosphere throughout the Arctic, an extensive validation of the algorithm is required. The validation is presented in Section 3 of this manuscript.

We agree with the referee that a short description of the algorithmic concepts used in AEROSNOW and its recent improvements, which include, the coupling of novel cloud identification scheme described in Jafariserajehlou et al. (2019), would also improve the quality of the manuscript and its suitability for publication in AMT.

We have expanded the AEROSNOW description in Section 2.1.2 in the revised manuscript to include additional information about the *mechanics* of the algorithm. Section 2.1.2 from line 91 to line 141 has been rewritten in the revised manuscript. We would like to humbly ask the reviewer to re-read Section 2.1.2 in the revised manuscript.

The discussion of the spatial distribution and seasonal behavior of the retrieved AOD dataset was introduced for a specific reason: Although measurements over the central Arctic are sparse and knowledge about them is also limited, we had expectations about these distributions. In this regard, the purpose of examining these distributions was to gain further confidence in this new dataset and to test the distributions with respect to our expectations. For example, the AEROSNOW observation of increased pan-Arctic AOD values during spring is a clear confirmation that Arctic haze events were well captured by this dataset. We will also address our motivation to discuss the distributions in the revised manuscript.

Therefore, this AMT manuscript aims to strengthen the confidence in the AEROSNOW approach. However, the actual use and corresponding geophysical analysis of the obtained results has been presented in our another manuscript: The preprint is available in Atmospheric Chemistry and Physics Discussions (ACPD) (https://doi.org/10.5194/egusphere-2023-730).

Considering the above and our envisaged adjustments and additions, we strongly believe that AMT is a suitable journal for this type of work and this manuscript.

The AOD data obtained by AEROSNOW is well validated with the ground-based AERONET data over the high Arctic stations, and the AERONET data are considered to be of high quality ground based observations. Data from campaigns and other ground-based measurements would also have been useful for comprehensive validation. However, to the best of our knowledge, no public dataset is available that provides sufficient spatial and temporal statistics for our study period from 2003 to 2011 particularly over the snow- and ice-covered regions of the high Arctic. Unfortunately, the most valuable recent field measurements do not fall within our study period (such as, MOSAiC in 2019-2020, ACLOUD/PASCAL in 2017 (Wendisch et al., 2023), PAMARCMIPs in 2018 and 2021 (Nakoudi et al., 2018; Ohata et al., 2021)).

**At the end of line 69, we propose to add:** After retrieval and validation, we discuss the distribution of AOD over Arctic snow and ice in spring and summer, since ground-based or space-borne observational data on AOD covering the entire high Arctic cryosphere are limited. In this regard, the purpose of examining these distributions is to gain further confidence in this new data set and to test the distributions with respect to our expectations. For example, we examine whether the AOD retrieved by AEROSNOW is able to capture the increased pan-Arctic distribution of AOD in spring compared to summer (Willis et al., 2018), which would be a clear confirmation of whether or not Arctic haze events are well captured by this dataset.

**Q2:** Several significant sources of uncertainty are mentioned in the article but not adequately addressed. Firstly, cloud contamination poses a major uncertainty source, requiring further examination. Additionally, the assumption of a fixed snow surface parameterization is mentioned but not sufficiently analyzed. The article should discuss the uncertainties associated with these assumptions and explain why they hold true for the study region.
**Response:** This question is addressed in the answer to *Q4 (for cloud)* and *Q5 (for NDSI)* of the comments by the referee below.

**Q3:** In line 94-95, the author claims that the cloud identification algorithm meets the requirements for high-latitude AOD studies, but this statement requires a citation to support it. Additionally, the phrase "a given sampling period" in line 96 needs to be clarified since a larger time window could introduce risks to this assumption.

**Response:** We agree with the reviewer and propose to change the lines as mentioned below to clarify the reviewer's question. We have followed Jafariserajehlou et al. (2019) who have proven that the AATSR-SLSTR cloud detection algorithm (ASCIA) meets the requirements for high latitude AOD studies. See also the answers to **Q4** for additional details. Cloud-free scenes are assumed to be unchanged or only slightly changed for a given sampling period. The sampling period used in this study is ±30 minutes, while cloudy or partly cloudy scenes have much greater spatial and temporal variability.

**At the end of line 95, we propose to add the citation:** The AATSR-SLSTR cloud identification algorithm (ASCIA) meets the requirements for the high-latitude AOD study and uses a dedicated set of thresholds for radiation and time-series measurements (Jafariserajehlou et al., 2019).

**Line 97, we propose to modify:** Cloud-free scenes are assumed to be unchanged or only slightly changed for a given sampling period. We set this value to ±30 min in this study (Jafariserajehlou et al., 2019), while cloudy or partly cloudy scenes exhibit much greater spatial and temporal variability.

**Q4:** Line 172 introduces the use of cloud fraction as a parameter in the quality flag, yet the uncertainty associated with cloud fraction over the Arctic region is not discussed. It would be valuable to explore whether the limited impact of cloud fraction is due to the large uncertainty associated with this parameter.

**Response:** The ASCIA data product (see above) was validated by comparison with independent observations, such as synoptic surface observations (SYNOP), AErosol RObotic NETwork (AERONET) data, and the following satellite products: (i) the ESA standard cloud product from the nadir cloud plume of AATSR L2; (ii) the product from a method based on a clear snow spectral shape developed at IUP Bremen (Istomina et al., 2010); and (iii) the Moderate Resolution Imaging Spectroradiometer (MODIS) products. Compared to the ground-based SYNOP measurements, ASCIA achieved promising agreement of more than 95% and 83% within ±2 and ±1 okta, respectively. In general, ASCIA shows better performance in identifying clouds over a ground scene observed at high latitudes than other algorithms applied to AATSR measurements.

**At the end of line 97, we propose to add:** The ASCIA cloud detection algorithm achieved promising agreement of more than 95% and 83% within ±2 and ±1 okta, respectively, compared to ground-based synoptic surface observations (SYNOP) (WMO, 1995). In general, ASCIA shows better performance in detecting clouds in a ground scene observed at high latitudes than other algorithms applied to AATSR measurements.

**Q5:** In line 118-119, the article mentions a fixed snow surface parameterization, the uncertainties related to this assumption should be analyzed and discussed. Specifically, it is important to explain why this assumption is valid for the study region, considering the Arctic's limited precipitation. Additionally, line 166 mentioned mixed snow regions, but its impact on AOD retrieval is not mentioned.

**Response:** To apply the fixed snow grain size approach and the assumption of snow contamination, we rely on the snow cover. Furthermore, the presence of snow in a pixel is defined by the Normalized Difference Snow Index (NDSI), an index that refers to the presence of snow in a pixel and is a more accurate description of snow detection compared to fractional snow cover (Riggs et al., 2017). Snow typically has a very high reflectance in the visible spectrum (VIS) and a very low reflectance in the shortwave infrared (SWIR). The NDSI is defined as the ratio of the difference

between the VIS and SWIR reflectance, i.e., NDSI = $((band_{vis} - band_{swir}) / (band_{vis} + band_{swir}))$. A pixel with an NDSI > 0.0 is considered to have snow cover, while a pixel with an NDSI <= 0.0 represents a snow-free land surface (Riggs et al., 2017).

In this study, the NDSI was used in a rigorous post-processing of the datasets to filter out the mixed and snow-free regions to minimize the impact of the surface on the top of atmospheric reflectance (TOA), therefore minimizing the impact on AOD retrieval.

Further, NDSI alone cannot describe the reflective properties of the surface. Therefore, the bidirectional snow reflectance distribution function (BRDF) model is also used. The BRDF model reproduces the directional variations in measured reflectance with an RMS error that is typically 0.005 in the visible wavelength range (Kokhanovsky and Breon, 2012), assuming a fixed snow grain size and snow contamination.

Please note that the retrieval is based on Equation 7 in the revised manuscript, which uses the ratio of simulated nadir BRDF values for the nadir and forward views of the dual-viewing instrument AATSR. With this strategy, we mitigate absolute errors in the BRDF but rather rely on the *shape of the BRDF* as seen from both directions. For our study, a narrow interval of NDSI was required to limit the BRDF-induced error in retrieving the AOD, which is less than 30% according to Istomina's approach (Istomina 2011, Section 3.3.3). Since we consider this error to be critical, we additionally introduced the Quality Flagging (QF) approach in our post-processing scheme, where we weighted the snow cover fraction even higher than the cloud fraction.

**At the end of line 119, we propose to add:** The BRDF model reproduces the directional variations in the measured reflectance with a root mean square (RMS) error that is typically 0.005 in the visible wavelength range (Kokhanovsky and Breon, 2012), assuming a fixed snow grain size and snow contamination. This assumption is valid because the model is also able to reproduce the directional signature of snow, although its directional signature is very different from that of other types of surfaces such as vegetation or bare soil.

**At the end of line 166, we propose to add:** The AEROSNOW retrieval is based on Equation 7 in the revised manuscript, which uses the ratio of the simulated nadir BRDF values for the nadir and forward views of the dual-viewing instrument AATSR. With this strategy, we mitigate the absolute errors in the BRDF but rather rely on the shape of the BRDF as seen from both directions. For our study, a narrow interval of NDSI was required to limit the BRDF-induced error in retrieving the AOD, which is less than 30% using Istomina's approach (Istomina 2011, Section 3.3.3). Since we consider this error to be critical, we additionally introduced the Quality Flagging (QF) approach in our post-processing scheme by adopting independent additional support from MODIS Terra and Aqua cloud fraction (Ackerman et. al., 2007) apart from ASCIA cloud detection algorithm, where we weighted the snow cover fraction even higher than the cloud fraction.

**Q6:** Some additional points to address include providing information about the basic aerosol properties of the models used (e.g., single scattering albedo and asymmetry factor) in line 137. Furthermore, the impact of a solar zenith angle cutoff of 75 degrees mentioned in line 160 should be discussed in terms of its impact on data sampling, particularly if there are specific times within a season when aerosol retrieval is not possible. Additionally, all monthly plots should display the variation in data for both AERONET and AEROSNOW datasets, including the number of retrieved data points aggregated into the monthly data. Lastly, in line 233, Figure 8 is introduced before Figure 7, which should be corrected.

**Response:** The basic aerosol properties used in the model, such as single scattering albedo (SSA), real part, and imaginary part of the refractive index for the coarse and accumulation modes of water-soluble, oceanic, dust, and soot aerosol components, are given in Table 1 in the revised

manuscript, which is adopted from the Istomina et al. (2011). Further, to avoid possible inaccuracies associated with particle shape, in this work on Arctic aerosol satellite retrieval we used the phase function for 550 nm measured ground-based by the Alfred Wegener Institute for Polar and Marine Research during one Arctic haze event on 23 March 2000 at Spitsbergen, Ny Ålesund, Svalbard, 78.923°, N 11.923° E, instead of the asymmetry factor.

We follow the recommendations in Istomina (Istomina 2011, Section 3.3.5) for a solar zenith angle (SZA) cutoff of 75 degrees based on sensitivity analysis using the SCIATRAN radiative transfer model (see also Mei et al., 2023). We agree with the referee that such a cutoff of the SZA leads to below average light conditions where the retrieval is working at the limit of its feasibility. We have done this deliberately, because March, being an important haze-event month, is also having such low light conditions and would have been excluded when we would have filterered more conservatively. Thus, keeping March in this dataset is an approach to extract as much as possible from the data to provide the most comprehensive view for the scientific community. March is already a time when Arctic haze events are relevant. Keeping March in this dataset was an attempt to extract as much as possible from the data to provide the most comprehensive view to the scientific community. A similar argument applies to summer, when persistent cloud cover also has a significant impact on the representativeness of the data. In our approach, we have applied a compromise between data yield and statistical representativeness.

We revised Fig.4: The revised presentation in monthly data is shown in the figure below and is also added in the revised manuscript of Fig.4.

[Figure]

*Figure 1: Figure 4. Validation of monthly mean AEROSNOW retrieved AOD colocated with monthly mean AERONET observation AOD obtained over PEARL, OPAL, Hornsund, and Thule stations. The linear regression lines are shown as blue dashed line and the bars are of one standard deviation.*

Yes we agree with the referee, we will cite both Figure 7 and 8 in line 233 of the revised manuscript.

**At the end of line 137, we propose to add:** The basic aerosol properties used in the model, such as the single scattering albedo (SSA), the real part, and the imaginary part of the refractive index for the coarse and accumulation modes of the water-soluble, oceanic, dust, and soot aerosol components are given in Table 1 adopted from Istomina et al. (2011). Further, in this work, we used the phase function for 550 nm measured ground-based by the Alfred Wegener Institute for Polar and Marine Research during the Arctic haze event on 23 March 2000 at Spitsbergen, Ny Ålesund, Svalbard, 78.923°, N 11.923° E.

**At the end of line 160, we propose to add:** In this work, we adopted the recommendations in Istomina (Istomina 2011, Section 3.3.5) for a solar zenith angle (SZA) of 75 degrees based on sensitivity analysis using the SCIATRAN radiative transfer model (see also Mei et al., 2023).

**We propose to change line 233:** However, comparing the seasonally averaged climatology from 2003 to 2011, the AEROSNOW results indicate higher AOD in the spring, and smaller values in summer shown in Fig. 7 and Fig. 8, which was expected due to the Arctic haze events (Willis et al., 2018).

**References:**

- Jafariserajehlou, S., Mei, L., Vountas, M., Rozanov, V., Burrows, J. P., and Hollmann, R.: A cloud identification algorithm over the Arctic for use with AATSR–SLSTR measurements, Atmospheric Measurement Techniques, 12, 1059–1076, 2019, https://doi.org/10.5194/amt-12-1059-2019

- Wendisch, M., et al. "Atmospheric and surface processes, and feedback mechanisms determining Arctic amplification: A review of first results and prospects of the (AC) 3 project." Bulletin of the American Meteorological Society 104.1 (2023): E208-E242.

- Ohata, S., Koike, M., Yoshida, A., Moteki, N., Adachi, K., Oshima, N., Matsui, H., Eppers, O., Bozem, H., Zanatta, M., et al.: Arctic black carbon during PAMARCMiP 2018 and previous aircraft experiments in spring, Atmospheric Chemistry and Physics, 21, 15 861–15 881, 2021, https://doi.org/10.5194/acp-21-15861-2021

- Nakoudi, K., Ritter, C., Neuber, R., and Müller, K. J.: Optical Properties of Arctic Aerosol during PAMARCMiP 2018, 2018. Nummelin, A., Li, C., and Hezel, P. J.: Connecting ocean heat transport changes from the midlatitudes to the Arctic Ocean, Geophysical Research Letters, 44, 1899–1908, 2017

- Riggs, G. A., Hall, D. K., and Román, M. O.: Overview of NASA's MODIS and visible infrared imaging radiometer suite (VIIRS) snow-cover earth system data records, Earth System Science Data, 9, 765–777, 2017. https://doi.org/10.5194/essd-9-765-2017

- Linlu Mei, Vladimir Rozanov, Alexei Rozanov, and John P. Burrows: SCIATRAN software package (V4.6): update and further development of aerosol, clouds, surface reflectance databases and models. GMD, 16, 1511–1536, 2023. https://doi.org/10.5194/gmd-16-1511-2023

- Ackerman, Steven A., et al. "Cloud detection with MODIS. Part II: validation." Journal of Atmospheric and Oceanic Technology 25.7 (2008): 1073-1086. DOI: 10.1175/2007JTECHA1053.

- Istomina, L., von Hoyningen-Huene, W., Kokhanovsky, A., and Burrows, J.: Retrieval of aerosol optical thickness in Arctic region using dual-view AATSR observations, in: Proc. ESA Atmospheric Science Conference, Barcelona, Spain, pp. 7–11, 2009.

- Istomina, L.: Retrieval of aerosol optical thickness over snow and ice surfaces in the Arctic using Advanced Along Track Scanning Radiometer, http://nbn-resolving.de/urn:nbn:de:gbv:46-00102463-15, 2011.

---

## Author Comment (AC2)

**Previous title:** Spring and summertime aerosol optical depth retrieval over the Arctic cryosphere by using satellite observations

**Revised title:** Retrieval of aerosol optical depth over the Arctic cryosphere during spring and summer using satellite observations: A validation and evaluation

The authors thank the referee for her/his effort, and time taken to review our manuscript. The valuable criticisms and comments have helped us to improve our paper. We hope that we have been able to answer satisfactorily the questions raised and clarify parts of the manuscript which were unclear or ambiguous.

We have changed the title of the manuscript to make it more interesting to AMT's readership.

In the following the referee comments and criticisms, our responses, as authors, and our resultant changes to the manuscript are colored black, blue and red respectively.

**Q1:** The title of the manuscript indicates some kind of development work regarding AOD retrievals over cryosphere. However, the manuscript is mainly about an analysis of AEROSNOW algorithm produced AOD from AATSR. The manuscript is basically lacking the development work that the title is suggesting. This makes me think if AMT is a suitable journal for this type of work.
**Response: [***We agree with the referee. The criticism expressed is also very similar to that of referee #1. We respond to both referees with the same text.***]

**Response:** We regret that the description in lines 65 to 69 of the manuscript did not convey to the referee clearly enough our objectives. Our main objective of the study described in this manuscript is to retrieve the aerosol optical depth (AOD) over the pan-Arctic cryosphere using a novel optimized algorithm, AEROSNOW, applied to measurements of the reflectance at the top of the atmosphere, measured by AATSR, made from the low earth orbit sun synchronous satellite Envisat from 2003 to 2011. Previous studies, retrieving AOD using the measurements of passive remote sensing instrumentation from space by others do typically not provide values over the pan-Arctic cryosphere region. So far, earlier versions of the retrieval were used to test the ability to retrieve AOD above Spitsbergen/Svalbard. In this manuscript, we describe an improved algorithm originally developed in house at the Institute of Environmental Physics at the University of Bremen and published by Istomina et al. (2009, 2011). As we have retrieved AOD over the cryosphere throughout the Arctic, an extensive validation of the algorithm is required. The validation is presented in Section 3 of this manuscript.

We agree with the referee that a short description of the algorithmic concepts used in AEROSNOW and its recent improvements, which include, the coupling of novel cloud identification scheme described in Jafariserajehlou et al. (2019), would also improve the quality of the manuscript and its suitability for publication in AMT.
We have expanded the AEROSNOW description in Section 2.1.2 in the revised manuscript to include additional information about the *mechanics* of the algorithm. Section 2.1.2 from line 91 to line 141 has been rewritten in the revised manuscript. We would like to humbly ask the reviewer to re-read Section 2.1.2 in the revised manuscript.

The discussion of the spatial distribution and seasonal behavior of the retrieved AOD dataset was introduced for a specific reason: Although measurements over the central Arctic are sparse and knowledge about them is also limited, we had expectations about these distributions. In this regard, the purpose of examining these distributions was to gain further confidence in this new dataset and to test the distributions with respect to our expectations. For example, the AEROSNOW observation of increased pan-Arctic AOD values during spring is a clear confirmation that Arctic haze events were well captured by this dataset. We will also address our motivation to discuss the distributions in the revised manuscript.

Therefore, this AMT manuscript aims to strengthen the confidence in the AEROSNOW approach. However, the actual use and corresponding geophysical analysis of the obtained results has been presented in our another manuscript: The preprint is available in Atmospheric Chemistry and Physics Discussions (ACPD) (https://doi.org/10.5194/egusphere-2023-730).

Considering the above and our envisaged adjustments and additions, we strongly believe that AMT is a suitable journal for this type of work and this manuscript.

The AOD data obtained by AEROSNOW is well validated with the ground-based AERONET data over the high Arctic stations, and the AERONET data are considered to be of high quality ground based observations. Data from campaigns and other ground-based measurements would also have been useful for comprehensive validation. However, to the best of our knowledge, no public dataset is available that provides sufficient spatial and temporal statistics for our study period from 2003 to 2011 particularly over the snow- and ice-covered regions of the high Arctic. Unfortunately, the most valuable recent field measurements do not fall within our study period (such as, MOSAiC in 2019-2020, ACLOUD/PASCAL in 2017 (Wendisch et al., 2023), PAMARCMIPs in 2018 and 2021 (Nakoudi et al., 2018; Ohata et al., 2021)).

**At the end of line 69, we propose to add:** After retrieval and validation, we discuss the distribution of AOD over Arctic snow and ice in spring and summer, since ground-based or space-borne observational data on AOD covering the entire high Arctic cryosphere are limited. In this regard, the purpose of examining these distributions is to gain further confidence in this new data set and to test the distributions with respect to our expectations. For example, we examine whether the AOD retrieved by AEROSNOW is able to capture the increased pan-Arctic distribution of AOD in spring compared to summer (Willis et al., 2018), which would be a clear confirmation of whether or not Arctic haze events are well captured by this dataset.

**Q2:** The analysis shown in the manuscript is based on AEROSNOW algorithm. The authors state that the AEROSNOW algorithm is based on Istomina et al. (2011) that has been further developed by the authors of this manuscript. There are no citations to any work that would describe the further developments by the authors neither any detailed descriptions is given in this manuscript. If this manuscript describes the further development, it is very much lacking the details necessary even to basic understanding on how the algorithm works. There are no details to reproduce the results used in AEROSNOW retrievals based on this manuscript or at least it would require an extensive literature search and details from multiple cited papers. This is a big issue in this manuscript.
**Response:** Yes, we agree with the referee that a brief repetition of the algorithmic concepts of AEROSNOW and its improvements, which include a novel cloud identification scheme by Jafariserajehlou et al. (2019), would also improve the quality of the manuscript and its suitability for publication in AMT.

As explained in our response to the question (**Q1**) we have expanded the AEROSNOW description in Section 2.1.2 in the revised manuscript to include additional information about the *mechanics* of the algorithm. Section 2.1.2 from line 91 to line 141 has been completely rewritten in the revised

manuscript. We would like to humbly ask the referee to re-read Section 2.1.2 in the revised manuscript.

Kindly note that retrieving AOD over a poorly understood complex system such as the Arctic with low-level clouds and very highly reflective snow-covered surfaces is a challenging task. Therefore, to tackle this AOD retrieval, we need to use different aspects of the concepts, especially from these three different works, such as the AEROSNOW algorithm by Istomina et al. (2009, 2011), the bidirectional snow reflectance distribution model (BRDF) by (Kokhanovsky and Breon, 2012), and the AATSR-SLSTR cloud identification algorithm (ASCIA) by Jafariserajehlou et al. (2019). Further, we have tried our best to put all the information in the revised version of the manuscript.

We have rewritten section 2.1.2 from line 91 to line 141 regarding the AEROSNOW algorithm in the revised manuscript and added all the information from the multiple cited papers for the reader.

**Q3:** The performance of the AEROSNOW show in the manuscript is in principle very good. I, however, feel there is a major issue in the analysis. The authors derive a quality flag (QF) parameter. The derivation and use of this parameter is not clear in the manuscript. As far as I understand both AERONET and AEROSNOW AOD are used in computing these results. Furthermore, the QF parameter is later applied to post-process filtering of the AEROSNOW data. This makes the AEROSNOW data dependent on AERONET and it is a major issue as no independent validation data is available. Then all the analysis is based on the dataset that has a dependency on AERONET data. This may result in overoptimistically good results. As mentioned, due to unclear description of the QF parameter I may have misunderstood this as well. In any case, the derivation and use of QF needs to be clarified and analysis of the results needs to be carried out using independent AERONET data.
**Response: [**_The first part of this answer is identical to the one to Referee #1 for the response to question (Q5)_**]**

To apply the fixed snow grain size approach and the assumption of snow contamination, we rely on the snow cover. Furthermore, the presence of snow in a pixel is defined by the Normalized Difference Snow Index (NDSI), an index that refers to the presence of snow in a pixel and is a more accurate description of snow detection compared to fractional snow cover (Riggs et al., 2017). Snow typically has a very high reflectance in the visible spectrum (VIS) and a very low reflectance in the shortwave infrared (SWIR). The NDSI is defined as the ratio of the difference between the VIS and SWIR reflectance, i.e., NDSI = (($band_{vis}$ - $band_{swir}$) / ($band_{vis}$ + $band_{swir}$)). A pixel with an NDSI > 0.0 is considered to have snow cover, while a pixel with an NDSI <= 0.0 represents a snow-free land surface (Riggs et al., 2017).

In this study, the NDSI was used in a rigorous post-processing of the datasets to filter out the mixed and snow-free regions to minimize the impact of the surface on the top of atmospheric reflectance (TOA), therefore minimizing the impact on AOD retrieval.

Further, NDSI alone cannot describe the reflective properties of the surface. Therefore, the bidirectional snow reflectance distribution function (BRDF) model is also used. The BRDF model reproduces the directional variations in measured reflectance with an RMS error that is typically 0.005 in the visible wavelength range (Kokhanovsky and Breon, 2012), assuming a fixed snow grain size and snow contamination.

Please note that the retrieval is based on Equation 7 in the revised manuscript, which uses the ratio of simulated nadir BRDF values for the nadir and forward views of the dual-viewing instrument AATSR. With this strategy, we mitigate absolute errors in the BRDF but rather rely on the *shape of the BRDF* as seen from both directions. For our study, a narrow interval of NDSI was required to

limit the BRDF-induced error in retrieving the AOD, which is less than 30% according to Istomina's approach (Istomina 2011, Section 3.3.3). Since we consider this error to be critical, we additionally introduced the Quality Flagging (QF) approach in our post-processing scheme, where we weighted the snow cover fraction even higher than the cloud fraction.

Furthermore, there is no dependence on AERONET data in the QF, rather we used AERONET data only for comparison with AEROSNOW AOD when the ratio of AEROSNOW and AERONET AOD values was greater than 1.6 to figure out the weighting of the influence of snow cover fraction and cloud fraction on the high AOD observed by AEROSNOW compared to AERONET, using independent additional support from MODIS Terra and Aqua cloud fraction products (Ackerman et. al, 2007), apart from the ASCIA cloud detection algorithm, which may have confused the referee. As discussed above, we will explain the need for deriving and using the QF post-processing approach in the revised manuscript.

**At the end of line 166, we propose to add:** The AEROSNOW retrieval is based on Equation 7 in the revised manuscript, which uses the ratio of the simulated nadir BRDF values for the nadir and forward views of the dual-viewing instrument AATSR. With this strategy, we mitigate the absolute errors in the BRDF but rather rely on the shape of the BRDF as seen from both directions. For our study, a narrow interval of NDSI was required to limit the BRDF-induced error in retrieving the AOD, which is less than 30% using Istomina's approach (Istomina 2011, Section 3.3.3). Since we consider this error to be critical, we additionally introduced the Quality Flagging (QF) approach in our post-processing scheme by adopting independent additional support from MODIS Terra and Aqua cloud fraction (Ackerman et. al., 2007) apart from ASCIA cloud detection algorithm, where we weighted the snow cover fraction even higher than the cloud fraction.

**More specific comments:**

**l.8** "The AOD is retrieved assuming that the surface reflectance observed by the satellite can be well-parametrized by a bidirectional snow reflectance distribution function, BRDF." More detailed analysis on the effects of this assumption needs to be carried out.
**Response:** [ The answer to this question is explained above in **Q3.** Kindly see the above answer]

**l.19** As Arctic is changing rapidly this almost 20 year old study already gives outdated information. More recent studies indicate AA of about 4 (e.g. Rantanen et al., "The Arctic has warmed nearly four times faster than the globe since 1979", Communications Earth & Environment, 3:168, 2022).
**Response:** We agree with the referee. We have changed the text and citation in our revised manuscript.

**At the end of line 19, we propose to add:** The Arctic has experienced a significant increase in near-surface air temperatures over the past three decades: the rate of temperature increase being about four times larger than the global mean (Rantanen et al., 2022). This phenomenon is known as Arctic Amplification (AA).

**l.104** "...an iterative procedure to obtain the AOD over Arctic snow and ice regions at 555 nm." No real details of the retrieval are given, e.g. what is the cost function in the iterative procedure.
**Response:** The cost function is defined by Equation 7 and minimized with respect to the aerosol optical depth (tau) in the revised manuscript, together with Equation 4 it defines the cost function. We will add at the end of the Section 2.1.2 the additional information that clarifies the cost function.

**At the end of Section 2.1.2, we propose to add:** The actual retrieval is working on Equations 7 and 4, and is minimized with respect to the optical depth (tau) at 555nm as a cost function.

**l.118** "We fixed the free parameters for the entire time series of AATSR, which involved a fixed snow grain size and snow impurity assumptions." How do these assumptions affect the retrievals?
**Response:** [ The answer to this question is explained above in **Q3.** Kindly see the above answer]

**l.161** Please define the NDSI.
**Response:** The Normalized Difference Snow Index (NDSI) is an index that refers to the presence of snow in a pixel and is a more accurate description of snow detection compared to fractional snow cover. Snow typically has a very high reflectance in the visible spectrum (VIS) and a very low reflectance in the shortwave infrared (SWIR). The NDSI is defined as the ratio of the difference between the VIS and SWIR reflectance, thus $NDSI = ((band_{vis} - band_{swir}) / (band_{vis} + band_{swir}))$ and is defined in Equation 8 of the revised manuscript.

The NDSI is defined in Equation 8 of the AEROSNOW algorithm in Section 2.1.2 of the revised manuscript.

**l.166** To better understand the effect of post-processing with NDSI, it would be nice to know how much of the data was filtered out and how much the metrics changed because of the filtering.
**Response:** Typically, we found bimodal distributions of AOD for the stations we considered, with the frequency of the second mode influenced by clouds and problematic surfaces. We filtered out 40% over problematic surfaces and residual clouds and 100% over central Greenland by using rigid post-processing with NDSI.

**At the end of line 166, we propose to add:** For the stations considered in this study, we found bimodal distributions of AOD, with the frequency of the second mode influenced by clouds and problematic surfaces. We filtered out 40% over problematic surfaces and residual clouds and 100% over central Greenland using rigid post-processing with NDSI.

**l.171** As mentioned earlier. The derivation and use of QF parameter needs to be clarified. Also independent validation data to validate AEROSNOW needs to be taken into account. Filtering cannot depend on AERONET data.
**Response:** [This question is similar to the question above in **Q3**. Kindly see the response to the question is explained above in **Q3**]

**l.195** "Uncertainties due to both space and time sampling differences are minimized." No details are given on how this is done and if the uncertainties really are minimized.
**Response:** The uncertainties due to spatial and temporal sampling differences are minimized according to the collocation strategy described in line 157 of the manuscript as "AERONET observations were averaged and compared to values measured within a 25-km radius and 30-minute time collocation of the AERONET stations for AEROSNOW".

**l.202** "...the latter does not take into account the relationships between geophysical variables." I do not believe this statement is true. Ordinary Least Squares regression is used exactly to determine the linear relationship between variables. Ordinary least squares does not take into account the uncertainties in x variable at all and, for example, therefore leads to biased slope estimates. Maybe this is what the authors meant here.
**Response:** The referee is right because these sentences are incorrect. The reduced major axis (RMA) regression takes into account the uncertainties (or errors) on the two variables, while the ordinary least squares (OLS) regression takes into account the uncertainties on one variable. Here we are looking for the best linear agreement between the two variables (which must be the same regardless of the variable in X or in Y, therefore we are looking for a symmetrical relationship). For this, different methods are possible (for example by minimizing the perpendicular distance or the triangle), and we choose the RMA (so by minimizing the triangle).

**The line 202, we propose to modify:** RMA regression takes into account the uncertainties or errors of the two variables, while ordinary least squares (OLS) regression takes into account the uncertainties of one variable. Here, the best linear fit between the two variables is sought, which must be the same in X or in Y regardless of the variable, that is, aiming for a symmetric relationship. For this, different methods are possible, for example, by minimizing the perpendicular distance or the triangle, and we choose the RMA, that is, by minimizing the triangle.

**l.203** "Therefore, it is recommended that the use of reduced major axis (RMA) regression replace the use of standard linear regression" RMA regression is not the only one tackling the issues of OLS. There are many other methods as well. This statement written by the authors may give the reader a wrong impression here.
**Response:** [The answer to this question is explained above in **l.202** question. Kindly see the above answer]

**l.218** "This difference can be explained by using an optimal chemical transport model by separating the total AOD to aerosol components." This is a vague statement not explained well enough and not supported by any results shown in this manuscript.
**Response:** Here we would like to cite our manuscript using AEROSNOW data to evaluate GEOS-Chem model, the preprint is available in Atmospheric Chemistry and Physics Discussions (ACPD) (https://doi.org/10.5194/egusphere-2023-730).

**At the end of line 218, we propose to add:** This difference can be explained by using a chemical transport model by separating the total AOD to aerosol components (Breider et. al., 2014, Swain et. al., 2023).

**l.246** The whole paragraph is a bit confusing. The paragraph is in the results section and no comparison between AEROSNOW and model data has been carried out.
**Response:** We propose to remove the paragraph at line 246 "The comparison of the AEROSNOW data with model data is the next logical step. We test the quality and limitations of the model as well as of the satellite dataset in the high Arctic. Further, the seasonal variability of AOD observed from AEROSNOW can then be attributed by investigating aerosol properties, components, long-range transport, and local sources. This is possible by using an optimal chemical transport model with updated emission inventories for this study period." in the revised manuscript.

We propose to remove the paragraph at line 246 in the revised manuscript.

**l.254** "The AEROSNOW algorithm uses the dual-viewing capability of the AATSR instrument to minimize retrieval uncertainties" It was not shown that the use of dual view instrument really minimizes the retrieval uncertainties.
**Response:** [The answer to this question is explained above in **Q3.** Kindly see the above answer]

Yes, dual-viewing capability of the AATSR instrument has already been used to minimize retrieval uncertainties in other works such as, Istomina et al., (2009, 2011), and Mei et. al., 2020.

**At the end of line 254, we propose to add:** The AEROSNOW algorithm uses the dual-viewing capability of the AATSR instrument to minimize retrieval uncertainties (Istomina et al., 2009, 2011, and Mei et. al., 2013b).

**l.255** "It showed good agreement with ground-based AERONET observations, with a correlation coefficient R = 0.86 and a low systematic bias." It may be totally misleading to give retrieval metrics without stating any details. Use of different time averaging for example hugely affects the metrics.

**Response:** We thank the referee this valuable comment, we agree with the referee that such a distinction is important. For this purpose we once performed the validation on daily basis and monthly validation. The daily validation shown in the Appendix Figure A1 of the manuscript and exhibit similar characteristic as the one for the monthly plot. The monthly mean values are derived from daily based investigations.

**l.256** "The high anthropogenic aerosol loading (Arctic haze events) due to long-range transport over Arctic snow and ice is captured by the AOD determined by AEROSNOW." There were no well justified experiments in the manuscript that would confirm this exact statement about long-range transport of anthropogenic aerosols. The authors themselves listed this type of experiment as future work.

**Response:** We would also like to cite our manuscript using AEROSNOW data to evaluate GEOS-Chem model, which is available as the preprint in Atmospheric Chemistry and Physics Discussions (ACPD) (https://doi.org/10.5194/egusphere-2023-730). Further, the monthly mean spatial maps shown in Fig.2 confirmed that the haze events are captured well.

**At the end of line 256, we propose to add:** The high anthropogenic aerosol loading (Arctic haze events) due to long-range transport (Willis et al., 2018) over Arctic snow and ice is captured by the AOD determined by AEROSNOW (see also Swain et. al., 2023). Further, the monthly mean spatial maps shown in Fig.2 confirmed that the haze events are captured well.

**l.263** "The promising AOD results obtained with AEROSNOW indicate that these can be used to evaluate and improve aerosol predictions for various chemical transport models." In previous parts of the manuscript, the authors say that in the future models can be used to test and validate the AEROSNOW retrievals. Here in the conclusions, the authors say that AEROSNOW can be used to evaluate and improve the models. This is confusing.

**Response:** We are sorry for the confusion. We consider the AOD data retrieved by AEROSNOW as a space-borne observational basis and therefore AEROSNOW can be used to evaluate state of the art models instead. Here it would make sense, that we cite our manuscript in which we use AEROSNOW data to evaluate the GEOS-Chem global chemical transport model. The preprint is available in Atmospheric Chemistry and Physics Discussions (ACPD). (https://doi.org/10.5194/egusphere-2023-730).

[revised manuscript text omitted]

---

## Referee Report (RR1)

The revised version of the paper significantly enhances the clarity of the fundamental structure of the research. It is now evident that the primary achievement of this study lies in the fusion of two previously unrelated algorithms into AEROSNOW, while incorporating quality control measures. This combination allows the algorithm's application for the reliable retrieval of aerosol optical depth (AOD) over the Arctic region. Once the outstanding issues are addressed, this paper appears suitable for publication in the Atmospheric Measurement Techniques (AMT) journal.

Nevertheless, certain areas still require clarification. In the prior review, several questions arose due to the reviewers' misunderstanding of the content. Although the author guided the reviewers to relevant sections within the article, it highlights a fundamental issue with the paper - its difficulty in comprehension. While all the necessary information may be present, it lacks a logical organization that would enable someone unfamiliar with the research to grasp and follow it seamlessly.

As a proposed improvement, restructuring the paper is recommended. Section 2 should exclusively focus on data, encompassing descriptions of all data used in the algorithm, such as the MODIS data employed in the quality assurance (QA) process. Section 3, dedicated to the algorithm, should begin with a clear presentation of the algorithm's flowchart, emphasizing that this paper's core contribution is the development of a robust aerosol retrieval algorithm over the Arctic. This is achieved by amalgamating two existing algorithms and implementing dependable QA procedures, expanding the algorithm's applicability across a wider region and time span. Sections 3.1 and 3.2 can address the pre-existing algorithms, while Section 3.3 should expound upon the novel elements (notably, the "new contribution" seems to occur in the post-processing phase).

In the introduction, it is imperative to elucidate the limitations of exclusively using Istomina, 2009 and underscore the key advantages of uniting these two algorithms. Distinguishing between a research algorithm tailored to specific cases and a robust algorithm capable of operating across an entire region and data records is pivotal. The paper should emphasize this crucial distinction to effectively persuade its readers.

I have some algorithm related questions as well, particularly concerning Istomina, 2009. Although this isn't the primary focus of your research, it's crucial due to its centrality in the algorithm. I'm not clear about where $\tau$ fits into the equations, which is the goal parameter. In my experience, $\tau$ should be in the $\rho_{atm}$ term in Eq. 3, which includes contributions from aerosol and Rayleigh and gas. Once you know how much is $\rho_{atm}$ you can get $\tau$ using an assumed aerosol model. However, between line 249 to line 267, you describe two different methods of estimate $\rho_{atm}$. Many questions regarding these two paragraphs. First, is this $\rho_{atm}$ the same in Eq. 4 vs. Eq. 3. If so, what is the point of having BRDF estimation if you already got $\rho_{atm}$. I assume this roughly estimated $\rho_{atm}$ is only for atmosphere correction to get a better BRDF ratio, which is used in iterative process to fine tune the $\rho_{atm}$ contribution. Eventually the $\rho_{atm}$ in Eq. 3 will be the same as $\rho_{atm}$ in Eq. 4.  Not sure if my understanding is correct. Because it is not clearly stated in the paper. Second, both methods of estimating $\rho_{atm}$ has their own

uncertainties, one assuming coastal aerosol properties are the same with inland, another uses pre-defined aerosol models, plus assuming negligible ocean surface contribution, which also raises big concerns. It is not clear whether any of the method considered sedimentation or other watercolor contribution.

Other clarification questions:

1. It is still not clear to me how can $\rho_{atm}$ converted to $\tau$ without an aerosol model (or maybe it is the same model stated in Table 1 and Figure 2?).
2. The airmass factor and Angstrom equation discussion from line 269 to line 275 does not provide sufficient connection to the previous text.
3. Eq. 18 is still very hard to understand because there is no specific definition of what $\rho$ is, on the second term in the righthand side. If $\rho_{sfc}$ is calculated from Eq. 15, then I assume $\rho$ is from observation. But based on which equation, is it Eq. 3?
4. Which the $\rho$ in Eq. 4 is the $\rho$ in Eq. 5? I assume is the left-hand side $\rho$. But when I read until Eq. 15, I realized the $\rho$ in Eq. 5 maybe $\rho_{sfc,sim}$. To enhance clarity, it would be helpful to move Eq. 15 to the beginning and explain how you solve Eq. 15 instead of Eq. 4.
5. Line 241 to 249 discuss uncertainties in Eq. 4, which again, doesn't align with your approach completely, so it requires further discussion.

Other points:

1. The abstract should explicitly state that AEROSNOW is a result of merging two existing algorithms.
2. The paragraph that commences with "Recently, Toth et al., (2018)" around line 65 lacks a clear connection with the preceding content. This connection should be established earlier in the paragraph.
3. The repetitive mention of "use AATSR to retrieve aerosols over the Arctic" should be minimized throughout the paper.
4. Lines 93 and 95 both refer to "cloud masking." It's important to clarify that these are distinct cloud masking procedures, and a detailed explanation is necessary. However, if the suggested structural changes are implemented, this may no longer be an issue.
5. Ensure proper citation is provided in the AATSR data section.
6. The paragraph discussing the lack of other field campaign validation data should be relocated from the AERONET data section to the introduction within the Data section.
7. In line 246, "but The ..." the capitalization of "T" in "The" is unnecessary.
8. Clarify whether this LUT is used for retrieval or solely for atmospheric correction in line 286.
9. Enhance the clarity of the flow chart by using color-coding to differentiate existing algorithms from new elements. The algorithm section should also encompass any additional pre-processing and post-processing steps applied, such as solar zenith angle (sza), snow-only filtering, and other quality control (QF) requirements.
10. Despite previous inquiries, there remains confusion regarding whether this algorithm is designed for partial snow cover or exclusively for 100% snow cover. Line 340 implies the consideration of snow cover fraction, suggesting applicability beyond 100% snow cover.

However, line 225 states that "only pure snow-covered areas (100% snow cover) are used." This discrepancy requires clarification.

11. Specify whether the seasonal mean is calculated from level 2 data or monthly mean data.

12. Figure 7 still lacks standard deviation information; consider including it for greater completeness.

---

## Author Response (AR2)

**Referee #1**

**Previous title:** Spring and summertime aerosol optical depth retrieval over the Arctic cryosphere by using satellite observations

**Revised title #2:** Retrieval of aerosol optical depth over the Arctic cryosphere during spring and summer using satellite observations

The authors would like to thank the reviewer for her/his interest and efforts to review our manuscript.

We hope that we have been able to answer satisfactorily the questions raised and clarify parts of the manuscript which were unclear or ambiguous.

We have changed the title of the manuscript to as per the first referee's suggestions.

In the following the referee comments and criticisms, our responses, as authors, and our resultant changes to the manuscript are colored black, blue and red respectively.

**Q1:** One of the main themes from the first-round review is whether or not this article fits in the scope of AMT. The author still didn't fully address this problem. AMT is meant for introducing new technique in measurements, if the article is only focus on validation and evaluation, plus analyses of aerosol distribution over the Arctic from satellite retrievals, there are many other articles out there for an article that is more "sciencey". However, in response to reviewer, the author emphasizes that there is new addition of algorithm development other than following Jafariserajehlou et al., (2019). If that is the case, then the paper is suitable for AMT, but the structure and the title need to reflect the content.

**Response:**

We would like to cite the aim and scope mentioned on the website of EGU/AMT journal as follows (last visited 08.09.2023): *"The main subject areas comprise the **development**, intercomparison, and **validation** of measurement instruments and techniques of data processing and information **retrieval** for gases, **aerosols**, and clouds."*.

For example, apart from other years, only in this year several articles were published with the specific focus on validation (such as Vinjamuri et. al., 2023; Baars et. al., 2023; Moallemi et. al., 2023; Gregor et. al., 2023; Ratynski et. al., 2023; Lerot et. al., 2023; Nyamsi et. al., 2023; Lange et. al., 2023; Garane et. al., 2023; Furlani et. al., 2023 etc.; see references at the end of this author's response).

*In this context, we would like to briefly repeat the strategy/content of our manuscript*:

> *Step 1: We presented an existing AOD retrieval algorithm (Istomina et al (2009)) published in 2009, which we improved by integrating a novel cloud identification scheme Jafariserajehlou et al., which is published a decade after in the year 2019.*

***Step 2:** We used the approach from step 1 and further introduced a post-processing scheme which additionally helped in filtering unreliable AOD retrievals.*

***Step 3:** Integrating step 1 and 2, and running the whole scheme for the period of AATSR observations led to significant improvements in the AOD retrieval. As a consequence, it was possible for the first time to run this integrated approach over a large part of the Arctic cryosphere. This prompted us to test the validity/quality on much larger scales than being done in the article of Istomina et al. (2009), leading to the results we have shown in our manuscript.*

We consider the integration of the individual steps as ***method development***. The method developed we called AEROSNOW. Consequently, a subsequent ***validation*** must show that the methodology delivers valid results. For this reason, we emphasize that this novel methodology and its test for validity fits very well with the aims and scope of the EGU/AMT journal.

In order to increase readability for the typical readership of the journal, we have made the changes in structure and the title of the revised manuscript as follows:

We have included the description of both large building blocks of the new scheme, i.e. Istomina et al., 2009 and Jafariserajehlou et al., 2019 into Section 2 "Data Sets and Algorithms" in the revised manuscript.

Further, we have added the description of the methodology into Section 3 "Methodology", in which we have specified the development of AEROSNOW scheme. For more clarity, we propose to add a flow chart of AEROSNOW in the revised manuscript. The flow chart of the building blocks of AEROSNOW is presented as Fig.1 below.

[Figure]

Fig.1 Flowchart describing the important building blocks of the AEROSNOW scheme.

**We propose to change the manuscript title as follows: "**Retrieval of aerosol optical depth over the Arctic cryosphere during spring and summer using satellite observations".

**Q2:** Assuming the latter scenario, here are the suggestions: 1. Move new development of AEROSNOW algorithm development out of section 2 and make it a new section. Because in Section 2 is meant to introduce already existing data and algorithm. Everything that is pre-existing can be in that section and the new addition should be in methodology or a section called algorithm new development. 2. Change the title. 3. Article spent a page introducing cloud mask. If it is from previous paper, the cloud mask introduction is needed, but can be summarized more concisely.
**Response:**
**Referee's suggestion 1 and 2:** We have followed the reviewer's recommendations and have made the structural changes that we outline in the text of our response to Q1.

**Referee's suggestion 3:** We have followed both of the reviewer's recommendations. Since the second reviewer suggests explaining the cloud masking approach (ASCIA) in more detail, we have summarized the cloud masking approach in Section 2.3.1 "Cloud detection Algorithm (ASCIA)".

**Q3 (i):** The second problem that still exist is the uncertainty estimates. Retrieving aerosol over ice/snow surface using passive sensor is extremely hard due to little information content within the TOA reflectance that is dominated by surface. Thus, a small error/assumption mismatch in surface will cause large error in aerosol retrieval. The article needs to show convincing results that the surface assumption is valid and give solid uncertainty estimation on surface introduced error. It is well known that snow grain size change drastically from fresh snow to aged snow to melting snow. The assumption assumes same snow grain size introduce large uncertainties over areas that new snow will deposit, or old snow will melt, which in Figure 2, the elevated aerosol near Greenland southern coasts, seems solid evidence of this type of error.
**Response:**
We thank the referee for acknowledging that the retrieval over bright surfaces is a difficult task.

We are aware that snow grain size variability, melt ponds, and impurities in the snow can cause problems. For this reason, we had to filter the data using the NDSI and apply the aforementioned post-processing. By taking these measures we were able to perform retrievals utilizing the (simple) BRDF parametrization according to Kokhanovsky and Breon (2012).

We believe that the outcome of our validation is precisely following your request for "convincing results that the surface assumption is valid".

The increased values of aerosols on the east coast of Greenland are located in the Fram Strait, which is characterized by strong ice export. We agree with the referee that this is a potentially problematic region, but for different reasons. The ice flows in this region might be in the size range of the instrumental spatial resolution and might not be correctly identified by the NDSI filter. Residual open ocean water fraction which is not identified by AEROSNOW might have an adverse effect on the retrieval quality.

However, there is also the possibility that the AOD values are accurate, as there is evidence that high wind speeds in this region can lead to sea salt emissions (Rhodes et al., 2017). For this reason, as suggested by the referee in Q1, we also compared the AEROSNOW data with the GEOS-Chem simulations (such as in AboEl-Fetouh et al., 2020/Hesaraki et. al., 2017, which is suggested by the referee as well in this review)

We found model-based evidence that high wind speeds increase the sea salt concentration in this area and might be responsible for increased AOD values (see Fig.3 below). In absence of high spatio-temporal validation data in this region, we can neither rule out that the data are problematic,

nor that the high values are induced by specific processes, such as high wind speeds. Consequently we want to keep these data points.

[Figure]

Fig.2 Mean climatological MAM and JJA AEROSNOW derived AOD over Arctic sea ice averaged from the year 2003 to 2011. White area shows lack of data apart from the masked land area.

[Figure]

Fig.3 Seasonal mean of Spring (MAM, top panel) and Summer (JJA, lower panel) of GEOS-Chem simulated total and speciated AOD over Arctic Sea Ice averaged from the year 2003 to 2011 colocated with AEROSNOW. The simulation of GEOS-Chem are done as per Hesaraki et. al., 2017.

**Q3 (ii):** The article says BRDF introduced error is 30%, but I don't know how that is calculated or estimated.
**Response:**
To explain the error caused by the BRDF, the analysis performed by Istomina (2011) (described in section 3.3.3 in her publicly available dissertation) is used. Fig.4 shows the relationship between the simulated AOD and the input AOD with the blue solid line for the unperturbed case and the colored areas for the reflectances perturbed by 5%. A higher reflectance leads to a higher AOD and vice versa. It can be seen that a constant perturbation of the reflections by 5% does not lead to an offset type retrieval error, but to an error that increases with AOD. This means that, as expected, the assumption of a small atmospheric influence on the TOA reflection does not hold for larger AODs. However, the retrieval error can be assumed to be constant for an AOD below 0.30 (Fig.4), given a value of about ± 0.1 for the absolute AOD error.

Furthermore, for the calculation of the relative error according to Istomina (2011), it can be assumed that the retrieval error is constant for the AOD below 0.3. However, in our study, one of the highest retrieved value of AOD is 0.15, so for example for an AOD value of 0.15 we obtain a simulated AOD value of 0.115 (Fig. 4), resulting in a relative error of 30%.

[Figure]

Figure 4: The dependence of the retrieved AOD on the input AOD for the RT-simulated TOA reflections, unperturbed (blue solid line), increased by 5% (upper edge of the color-filled area), decreased by 5% (lower edge of the color-filled area). Adopted from (Istomina (2011), section 3.3.3).

**At line 246, we propose to modify:**
For our study, a narrow interval of NDSI was required to limit the BRDF-induced error in retrieving the AOD, which is less than 30% using Istomina's approach (Istomina (2011), Section 3.3.3; Kokhanovsky and Breon, 2012).

**Q3 (iii):** The paper mentions that "The BRDF model is analytical and was compared with a series of multispectral and multidirectional measurements from the POLDER-3" However, it is not clear how are POLDER-3 TOA reflectance are guaranteed to be aerosol/thin clouds free. Also what is the result of the comparisons?
**Response:**
We apologize if the message was not clearly conveyed to the referee. The BRDF model used in the aerosol retrieval algorithm of Istomina et al. (2009) is from Kokhanovsky and Breon (2012). In their paper the authors validated the BRDF model with a series of multispectral and multidirectional measurements from POLDER-3. The authors ensured cloud-free conditions and verified the performance of the snow BRDF which was derived by a radiative transfer model for snow based on asymptotic solution. Such an approach is valid for a single scattering albedo of snow close to one (Kokhanovsky et al., 2011). Their results were validated using satellite measurements (POLDER) over the Arctic and Antarctic. It was found that the model can indeed be used to describe both the spectral and directional signatures of targets that are completely covered by snow. The deviations of the model from the measurements are less than 10%, and the correlation coefficients are above 0.85 in most cases analyzed over Arctic and Antarctica, although the model results are somewhat less anisotropic than the observations, especially in the forward scattering direction at large zenith viewing angles.

**Q4:** The third is that due to the limited AERONET site and their location, it is highly recommended author to compare the satellite retrieved AOD against field campaign results (e.g. POLAR-AOD, MOSAiC-ACA, and AFLUX - Arctic). Another points regarding using AERONET is that the uncertainty of AERONET AOD retrieval and aerosol property retrieval over polar region should be mentioned and discussed (Mazzola et al., 2012).
**Response:**
The valuable field campaigns mentioned by the referee such as POLAR-AOD, MOSAiC-ACA, and AFLUX-Arctic do not fall within our study period of 2003 to 2011, as follows:

**1. POLAR-AOD:**

- *The first POLAR-AOD campaign:* was performed at Ny-Ålesund (78 550 N, 11 560 E, Svalbard Islands, Norway) from **March 25 to April 5, 2006 (11 days),** near the Rabben station of the National Institute of Polar Research (NIPR, Japan). It was organized and hosted by the AWI German station.

  There are no colocations within 11 days measurement time period (**March 25th to April 5th, 2006)** (Mazzola et. al., 2011).

- The second POLAR-AOD campaign which was held for intercomparison with the first one was performed over canary islands at 28°N latitude is thus not suitable. (Mazzola et. al., 2011).

**2. MOSAiC-ACA:**

- MOSAiC ACA took place from **28th August until 16th September 2020** (Mech et. al., 2022).

**2. AFLUX/PASCAL – Arctic:**

- AFLUX – Arctic airborne campaign took from March 20th to April 15th 2019, close to th Svalbard islands (Mech et. al., 2022).

- PASCAL– Arctic shipborne campaign took from March 20th to April 15th 2019, close to th Svalbard islands (Wendisch et. al., 2019).

For all campaigns/expeditions after 2011 we could not achieve temporal collocation.

Yes, we agree with the referee, the uncertainty of AERONET AOD retrieval over the polar regions has been discussed as per Mazzola et al. (2012) in the revised manuscript.
**At Line 223 we propose to add:**
The use of data sources other than AERONET for validation would have been very helpful. Unfortunately, data from valuable campaigns and expeditions such as POLAR-AOD (Mazzola et al., 2011), MOSAiC Expedition and MOSAiC-ACA (Mech et al., 2022), and AFLUX/PASCAL - Arctic (Mech et al., 2022) were only available after 2011. For this reason, we have focused on validation with ground-based AERONET measurements.

There are some other points in the paper that need to be addressed:

1. The aerosol model provide lack of size distribution, also the parameter is not wavelength dependent, leaving question whether the retrieval is a single wavelength retrieval or utilizes more wavelength. Also there are many other aerosol model study over the study region that provide more comprehensive dataset (e.g. AboEl-Fetouh et al., 2020).
**Response:**
In this study, we used 555nm for the retrieval of the AOD.

We would like to bring to the attention of the referee that the GEOS-Chem model simulation results used in AboEl-Fetouh et al., 2020 are taken from Hesaraki et. al., 2017. These research articles focused only on the Canadian Arctic region for the time period of 2009 to 2012 for 4 years. Further,

in these two studies the focus was to compare GEOS-Chem simulated AOD with the AERONET retrieved AOD over the Canadian Arctic AERONET stations (such as Barrow, PEARL, OPAL, Resolute Bay, and Thule).

In our investigation, we have comprehensively examined a time frame spanning almost a decade, from 2003 to 2011, which involved extensive coverage of spatio-temporal central Arctic sea ice. In addition, we have incorporated data from the Hornsund AERONET station in our validation, in addition to the Canadian Arctic AERONET stations like PEARL, OPAL and Thule.

We agree with the referee that there are several model studies conducted over the Arctic. Mainly with the focus on understanding  the uncertainties resulting from the direct effects and those on clouds, which are poorly represented in models (Boucher et al., 2013). The deficiency of many models is particularly true for the central Arctic sea ice region due to the lack of datasets with high spatiotemporal observational coverage and understanding of aerosols in this region (Schmale et al., 2021). It is worth noting that several valuable studies have been conducted for Arctic aerosols using models (such as Hardenberg et. al., 2012; Evangeliou et. al., 2016; Sand et. al., 2017; Breider et. al., 2017; Ren et. al, 2020; Schmale et. al., 2021). All of these important and valuable studies cannot use aerosol observations over the entire central Arctic cryosphere with high spatial and temporal coverage.

We therefore think, that the datasets generated in our study will help to improve model results related to aerosol loading in the high Arctic. We hope that they will also be used  when creating new assimilated datasets in the central Arctic cryospheric region during ongoing Arctic warming.

*The sparseness of available AOD datasets derived from observations may explain, at least in part, the variations of AOD simulations from different climate models (Sand et al., 2017). Without doubt, the lack of high spatial-temporally representative AOD measurements in the Arctic limits our knowledge about radiative forcing and the Arctic warming in global and regional climate models (Goosse et al., 2018).*

2. What is the temporal resolution of the pre-constructed surface?
**Response:**
We have not used a pre-constructed surface, rather we used the BRDF model that reproduces the directional variations in the measured reflectance  (Kokhanovsky and Breon, 2012), assuming a fixed snow grain size and snow contamination. This assumption is valid because the model is also able to reproduce the directional signature of snow, although its directional signature is very different from that of other types of surfaces such as vegetation or bare soil. A snow BRDF model was selected for our study and is explained in Eq. 3 to Eq. 10 of the revised manuscript.

3. It is not clear that whether the retrieval is done only on pure snow-covered area (e.g. 100% snow cover) vs. the percentage of snow is estimated using NDSI.
**Response:**
We would like to refer the referee to the line 189 of the manuscript as "Some corrections are needed in order to take the snow structure into account as only pure snow-covered area is used for the retrieval and in the real AATSR measurements."

4. How are the data coverage impacted by solar zenith angle 75 degree cut off seasonally and spatially?

**Response:**
The effects of the solar zenith angle of 75 degrees cut off is shown in Fig. 2 of the manuscript. Fig. 2 of the manuscript is attached as Fig.5 of this author's response.

Because of the cutoff of the zenith angle of 75 degrees, aerosol retrieval coverage in March and October is spatially limited, as shown in Fig.2 of the manuscript as well as the Fig.5 of this author's response.

[Figure]

Fig.5. Pan-Arctic seasonal view of AEROSNOW retrieved AOD over snow and ice averaged from the year 2003 to 2011 for the months March to October, thus large parts of the period of insolation. Red circles indicate the location of AERONET stations

5. Paragraph 235 doesn't make sense and maybe better to place it where NDSI is introduced.
**Response:** We agree with the referee, we have moved the paragraph at line number 235 to 238 to line number 200 below the Eq.8.
**We propose to move the paragraph at line 235 to line 200:**
The NDSI is an index that refers to the presence of snow in a pixel and is a more accurate description of snow detection compared to fractional snow cover. Snow typically has a very high reflectance in the visible spectrum (VIS) and a very low reflectance in the shortwave infrared (SWIR). Snow coverage is determined by the NDSI ratio of the difference between the VIS and SWIR reflectance, and is defined in Eq. 13.

6. The filter of NDSI< 0.97 that is used to filter out 100% central Greenland isn't clear. Is there any reason for picking this 0.97? What does snow coverage looks like with different NDSI values? With that NDSI adjustment already included in SCF, is such a high value cut off using NDSI justified?
Response: *[The response to this question was described in Q5 in the authors response to referee #1 in our first round of answers and to Q3 in this round].*

To apply the approach at fixed snow grain size and account for potential snow contamination, we depend on snow cover. Moreover, the occurrence of snow in a pixel is established by NDSI, which offers a more precise depiction of snow detection as opposed to fractional snow cover (Riggs et al., 2017). Snow generally reflects the visible spectrum (VIS) quite well and exhibits low reflectance levels.  Reflectance in the shortwave infrared (SWIR). The NDSI is calculated as the ratio of the difference between the visible (VIS) and SWIR reflectance: NDSI = ((bandvis - bandswir) / (bandvis + bandswir)). A pixel exhibiting an NDSI > 0.0 is indicative of snow cover, while a pixel

with an NDSI <= 0.0 represents a snow-free ground surface (Riggs et al., 2017). The determination of the seemingly arbitrary threshold value of 0.97 was based on the validation process, which revealed clear over- or underestimations of the AOD when located over non-adequate surfaces.

In this study, the NDSI was used in a rigorous post-processing scheme to filter out the mixed and snow-free regions to minimize the impact of the surface on the top of atmospheric reflectance (TOA), therefore minimizing the impact on AOD retrieval. Fig.6 of this author's response shows the spatial snow cover over the Arctic as different NDSI values.

[Figure]

Fig.6. Pan-Arctic seasonal view of NDSI averaged from the year 2003 to 2011 for the months March to October, thus large parts of the period of insolation. Red circles indicate the location of AERONET stations.

7. Line 247 mentioned 30% less of BRDF-introduced error. The way this number is calculated isn't clear.
**Response:** *[We would ask the referee to read the author's answer to question Q3(ii), as we have already answered this question.]*

8. The uncertainty of MODIS cloud fraction over Arctic is not mentioned. It is expected to have large error (at least 10%) of CF over this region, with mostly missing thin cloud and cloud very close to the ground. How are these uncertainties impact the retrieval?
**Response:**
We agree with the referee. The MODIS CF products discrepancies with respect to Arctic has been discussed in the revised manuscript.

It is true that the CF from MODIS over the Arctic does not have the quality it usually has (Liu et al., 2022). However, it must be noted that this is part of the post-processing. A relatively reliable cloud masking has already been done using ASCIA. The post-processing shall be considered merely as an additional safeguard for the data. If it fails it induces additional loss of the data or can (also) not mask clouds (as ASCIA in one of the previous steps).
**At line 250, we propose to add:**
As per Liu et al.(2022) for the MODIS Terra and Aqua CF products over the Arctic, the discrepancies caused by different sensors and different algorithms are ±2% and ±5% with respect to the International Arctic Atmospheric Observing Systems (IASOA). The exact locations of the IASOA observatories are shown in Fig. 1 of (Uttal et al., 2016).

9. Figure 2 shall include another map of number of data points.
**Response:**
We agree with the referee. We have added the map of number of data points in the revised manuscript.

[Figure]

Fig.7. Pan-Arctic seasonal view of number of pixels used for calculating average AOD, averaged from the year 2003 to 2011 for the months March to October, thus large parts of the period of insolation. Red circles indicate the location of AERONET stations.

We propose to add map of number of data points presented above in the Fig.7 (in the manuscript as Fig. 4b) of the manuscript.

**At line 267, we propose to modify:**

The spatio-temporal frequency of observations over the Arctic from both ground and satellite is greater in summer (JJA) than in spring (MAM). Fig. 4(a) shows the monthly averaged AOD over snow and Arctic ice for the period 2003-2011, with significant differences in the spatial distribution of AOD. Fig. 4(b) shows the number of pixels used to average AOD per grid cell during 2003-2011 for March through October.

10. Is the reduced major axis method results significantly differ from normal linear fit? Although there is measurement error in AERONET, compared to satellite retrieved AOD, the measurement error in AERONET is almost negligible. Thus, the regular linear fit is also appropriate.
**Response:** *[A similar question has been raised by the 2nd referee in the first round of review. We post here our answer.]*

The reduced major axis (RMA) regression takes into account the uncertainties (or errors) on the two variables, while the ordinary least squares (OLS) regression takes into account the uncertainties on one variable. Here we are looking for the best linear agreement between the two variables (which must be the same regardless of the variable in X or in Y, therefore we are looking for a symmetrical relationship). For this, different methods are possible (for example by minimizing the perpendicular distance or the triangle), and we choose the RMA (so by minimizing the triangle).

This can be seen in the validation Figure 8 and 9 given below. For Hornsund and Thule AERONET station, the slope has increased from 0.71 and 0.88 in linear fit approach to 0.88, 1.01 in reduced major axis approach respectively.

[Figure]

Fig.8. Validation of monthly mean AEROSNOW retrieved AOD colocated with monthly mean AERONET observation AOD obtained over PEARL, OPAL, Hornsund, and Thule stations. The linear regression lines are shown as blue dashed line (for reduced major axis approach)

[Figure]

Fig.9. Validation of monthly mean AEROSNOW retrieved AOD colocated with monthly mean AERONET observation AOD obtained over PEARL, OPAL, Hornsund, and Thule stations. The linear regression lines are shown as blue dashed line (for normal linear fit approach).

11. Figure 6, the elevated aerosol loading in April is not shown within these three circles where AERONET located at. Elevated AOD exist at outside rim and there is no data to validate that.
**Response:**
There is a misunderstanding here. Unfortunately, we failed to point out that the red circles in Fig. 2 do not represent the radius of the collocation of 25km (they would have been much too small for visualization on the pan-Arctic map), but are only for orientation. To avoid such misunderstandings, we have added a note to the caption of Figure 2.
**We propose to add in the title of the Fig.2:**
Pan-Arctic seasonal view of AEROSNOW retrieved AOD over snow and ice averaged from the year 2003 to 2011 for the months March to October, thus large parts of the period of insolation. Red

circles indicate the location of AERONET stations for guidance. The size of the red circles is not identical to the spatial collocation radius of 25 km, which we have used in the validation.

12. Line 314-323 that section that explains the identification of aerosol over sea ice should be in methodology.
**Response:**
We agree with the referee.
We propose to move the paragraph at line 314-323 to section 2.3.2 of the revised manuscript.

**References:**

- Vinjamuri, K. S., Vountas, M., Lelli, L., Stengel, M., Shupe, M. D., Ebell, K., & Burrows, J. P. (2023). Validation of the Cloud_CCI (Cloud Climate Change Initiative) cloud products in the Arctic. Atmospheric Measurement Techniques, 16(11), 2903-2918. https://doi.org/10.5194/amt-16-2903-2023.

- Baars, Holger, Joshua Walchester, Elizaveta Basharova, Henriette Gebauer, Martin Radenz, Johannes Bühl, Boris Barja, Ulla Wandinger, and Patric Seifert. "Long-term validation of Aeolus L2B wind products at Punta Arenas, Chile and Leipzig, Germany." Atmospheric Measurement Techniques Discussions 2022 (2022): 1-34. https://doi.org/10.5194/amt-16-3809-2023.

- Moallemi, Alireza, Robin Lewis Modini, Benjamin Tobias Brem, Barbara Bertozzi, Philippe Giaccari, and Martin Gysel-Beer. "Concept, absolute calibration and validation of a new, bench-top laser imaging polar nephelometer." EGUsphere (2023): 1-39. https://doi.org/10.5194/egusphere-2023-392.

- Gregor, Philipp, Tobias Zinner, Fabian Jakub, and Bernhard Mayer. "Validation of a camera-based intra-hour irradiance nowcasting model using synthetic cloud data." Atmospheric Measurement Techniques Discussions 2023 (2023): 1-24. https://doi.org/10.5194/amt-16-3257-2023.

- Ratynski, Mathieu, Sergey Khaykin, Alain Hauchecorne, Robin Wing, Jean-Pierre Cammas, Yann Hello, and Philippe Keckhut. "Validation of Aeolus wind profiles using ground-based lidar and radiosonde observations at Réunion island and the Observatoire de Haute-Provence." Atmospheric Measurement Techniques 16, no. 4 (2023): 997-1016. https://doi.org/10.5194/amt-16-997-2023.

- Wandji Nyamsi, William, Yves-Marie Saint-Drenan, Antti Arola, and Lucien Wald. "Further validation of the estimates of the downwelling solar radiation at ground level in cloud-free conditions provided by the McClear service: the case of Sub-Saharan Africa and the Maldives Archipelago." Atmospheric Measurement Techniques 16, no. 7 (2023): 2001-2036.https://doi.org/10.5194/amt-16-2001-2023.

- Lange, Kezia, Andreas Richter, Anja Schönhardt, Andreas C. Meier, Tim Bösch, André Seyler, Kai Krause et al. "Validation of Sentinel-5P TROPOMI tropospheric NO 2 products by comparison with NO 2 measurements from airborne imaging DOAS, ground-based stationary DOAS, and mobile car DOAS measurements during the S5P-VAL-DE-Ruhr

campaign." Atmospheric Measurement Techniques 16, no. 5 (2023): 1357-1389. https://doi.org/10.5194/amt-16-1357-2023.

- Garane, Katerina, Ka Lok Chan, Maria-Elissavet Koukouli, Diego Loyola, and Dimitris Balis. "TROPOMI/S5P Total Column Water Vapor validation against AERONET ground-based measurements." Atmospheric Measurement Techniques 16, no. 1 (2023): 57-74. https://doi.org/10.5194/amt-16-57-2023.

- Furlani, Teles C., RenXi Ye, Jordan Stewart, Leigh R. Crilley, Peter M. Edwards, Tara F. Kahan, and Cora J. Young. "Development and validation of a new in situ technique to measure total gaseous chlorine in air." Atmospheric Measurement Techniques 16, no. 1 (2023): 181-193. https://doi.org/10.5194/amt-16-181-2023.

- Hesaraki, Sareh, Norman T. O'Neill, Glen Lesins, Auromeet Saha, Randall V. Martin, Vitali E. Fioletov, Konstantin Baibakov, and Ihab Abboud. "Comparisons of a chemical transport model with a four-year (April to September) analysis of fine-and coarse-mode aerosol optical depth retrievals over the Canadian Arctic." Atmosphere-Ocean 55, no. 4-5 (2017): 213-229. https://doi.org/10.1080/07055900.2017.1356263.

- Mech, Mario, André Ehrlich, Andreas Herber, Christof Lüpkes, Manfred Wendisch, Sebastian Becker, Yvonne Boose et al. "MOSAiC-ACA and AFLUX-Arctic airborne campaigns characterizing the exit area of MOSAiC." Scientific data 9, no. 1 (2022): 790. https://www.nature.com/articles/s41597-022-01900-7.

- Liu, Xinyan, Tao He, Lin Sun, Xiongxin Xiao, Shunlin Liang, and Siwei Li. "Analysis of Daytime Cloud Fraction Spatiotemporal Variation over the Arctic from 2000 to 2019 from Multiple Satellite Products." Journal of Climate 35, no. 23 (2022): 7595-7623. https://doi.org/10.1175/JCLI-D-22-0007.1.

- Uttal, Taneil, Sandra Starkweather, James R. Drummond, Timo Vihma, Alexander P. Makshtas, Lisa S. Darby, John F. Burkhart et al. "International Arctic systems for observing the atmosphere: an international polar year legacy consortium." Bulletin of the American Meteorological Society 97, no. 6 (2016): 1033-1056. https://doi.org/10.1175/BAMS-D-14-00145.1.

- Schmale, Julia, Paul Zieger, and Annica ML Ekman. "Aerosols in current and future Arctic climate." Nature Climate Change 11, no. 2 (2021): 95-105. https://www.nature.com/articles/s41558-020-00969-5.

- Von Hardenberg, J., L. Vozella, C. Tomasi, V. Vitale, A. Lupi, M. Mazzola, T. P. C. Van Noije, A. Strunk, and A. Provenzale. "Aerosol optical depth over the Arctic: a comparison of ECHAM-HAM and TM5 with ground-based, satellite and reanalysis data." Atmospheric Chemistry and Physics 12, no. 15 (2012): 6953-6967. https://doi.org/10.5194/acp-12-6953-2012.

- Evangeliou, Nikolaos, Yves Balkanski, Wei Min Hao, Alexander Petkov, Robin P. Silverstein, Rachel Corley, Bryce L. Nordgren et al. "Wildfires in northern Eurasia affect the budget of black carbon in the Arctic–a 12-year retrospective synopsis (2002–2013)." Atmospheric Chemistry and Physics 16, no. 12 (2016): 7587-7604. https://doi.org/10.5194/acp-16-7587-2016.

- Sand, Maria, Bjørn H. Samset, Yves Balkanski, Susanne Bauer, Nicolas Bellouin, Terje K. Berntsen, Huisheng Bian et al. "Aerosols at the poles: an AeroCom Phase II multi-model evaluation." Atmospheric Chemistry and Physics 17, no. 19 (2017): 12197-12218. https://doi.org/10.5194/acp-17-12197-2017.

- Breider, Thomas J., Loretta J. Mickley, Daniel J. Jacob, Cui Ge, Jun Wang, Melissa Payer Sulprizio, Betty Croft et al. "Multidecadal trends in aerosol radiative forcing over the Arctic: Contribution of changes in anthropogenic aerosol to Arctic warming since 1980." Journal of Geophysical Research: Atmospheres 122, no. 6 (2017): 3573-3594. https://doi.org/10.1002/2016JD025321.

- Ren, Lili, Yang Yang, Hailong Wang, Rudong Zhang, Pinya Wang, and Hong Liao. "Source attribution of Arctic black carbon and sulfate aerosols and associated Arctic surface warming during 1980–2018." Atmospheric Chemistry and Physics 20, no. 14 (2020): 9067-9085. https://doi.org/10.5194/acp-20-9067-2020.

- Rhodes, Rachael H., Xin Yang, Eric W. Wolff, Joseph R. McConnell, and Markus M. Frey. "Sea ice as a source of sea salt aerosol to Greenland ice cores: a model-based study." Atmospheric Chemistry and Physics 17, no. 15 (2017): 9417-9433. https://doi.org/10.5194/acp-17-9417-2017

-

**Referee #2**

**Previous title:** Spring and summertime aerosol optical depth retrieval over the Arctic cryosphere by using satellite observations

**Revised title #2:** Retrieval of aerosol optical depth over the Arctic cryosphere during spring and summer using satellite observations

The authors would like to thank the reviewer for her/his interest and efforts to review our manuscript.

We hope that we have been able to answer satisfactorily the questions raised and clarify parts of the manuscript which were unclear or ambiguous.

We have changed the title of the manuscript to as per the first referee's suggestions.

In the following the referee comments and criticisms, our responses, as authors, and our resultant changes to the manuscript are colored black, blue and red respectively.

I appreciate the effort of the authors in addressing my previous comments. However, in my opinion, the manuscript still has significant issues.

**Q1:** I asked the authors to give more actual details about the algorithm that was developed. The authors have added more text to the section 2.1.2. but many of the additions are just some results not really suitable for the method development part of the manuscript and citations without any details given. For example: "We set this value to ±30 min in this study (Jafariserajehlou et al., 2019), while cloudy or partly cloudy scenes exhibit much greater spatial and temporal variability." Without good knowledge on work by Jafariserajehlou et al. I do not understand what is meant here. The authors also give some results with no further details, for example on what region/stations and time were used. An example: "The ASCIA cloud detection algorithm achieved promising agreement of more than 95% and 83% within ±2 and ±1 okta, respectively, compared to ground-based synoptic surface observations (SYNOP) (WMO, 1995)".
**Response:**
Your criticism and that of referee 1 led us to thoroughly revise the description of the methods section of this manuscript. For a better understanding, we briefly repeat the building blocks of our method (same text as the answer to Referee 1):

> *Step 1: We presented an existing AOD retrieval algorithm (Istomina et al (2009)) published in 2009, which we improved by integrating a novel cloud identification scheme Jafariserajehlou et al. (2019), which is published in 2019. The cloud detection algorithm is presented in Jafariserajehlou et al. (2019).*
>
> *Step 2: We used the approach from step 1 and further introduced a post-processing scheme which additionally helped in filtering unreliable AOD retrievals.*
>
> *Step 3: By integrating step 1 and 2 and running the whole scheme for the period of AATSR*

*observations led to significant improvements in the AOD retrieval. As a consequence, it was possible for the first time to run this integrated approach over a large part of the Arctic cryosphere. This prompted us to test the validity/quality on much larger scales than being done in the article of Istomina et al. (2009), leading to the results we have shown in our manuscript.*

For more clarity, we propose to add a flow chart of AEROSNOW in the revised manuscript as follows:

[Figure]

Fig.1 Flowchart describing the important building blocks of the AEROSNOW scheme.

More details on cloud masking (ASCIA) and the AOD retrieval are given in dedicated subsections of section 2.3. The integration of the individual building blocks (as illustrated in the flowchart shown above) is described in a new section 3 "Methodology".

*Regarding your question concerning ASCIA:* In this study, cloud-free scenes are assumed to be unchanged or only slightly changed for a given sampling period of ±30 min, while cloudy or partly cloudy scenes exhibit much greater spatial and temporal variability as per (Jafariserajehlou et al., 2019).

The cloud fraction retrieved by ASCIA has been validated with SYNOP (WMO, 1995) ground-based cloud fraction measurements for the time period of AATSR (the year 2003 to 2011) and

SLSTR (the year 2018). The World Meteorological Organization (WMO) established SYNOP stations for weather information around the world. The locations of the SYNOP stations used for the ASCIA validation within the Arctic circle are shown in Fig. 1 of this author's response, as well as added below.

[Figure]

**Fig.1** SYNOP network coverage over the Arctic, the dark-blue points indicate the location of SYNOP stations (adopted from Jafariserajehlou et al. (2019)).
**We propose to change the manuscript title as follows: "**Retrieval of aerosol optical depth over the Arctic cryosphere during spring and summer using satellite observations".

**Q2:** The authors also give some statements that are not justified in any way and therefore cannot be verified. Example: "In general, ASCIA shows better performance in detecting clouds in a ground scene observed at high latitudes than other algorithms applied to AATSR measurements."
**Response:**
We refer to the assessment performed in Jafariserajehlou et al. (2019) where the ASCIA cloud detection algorithm achieved promising agreement of more than 95% and 83% within ±2 and ±1 okta, respectively, by comparison with ground-based synoptic surface observations (SYNOP) (WMO, 1995). In general, ASCIA shows better performance in detecting clouds in a ground scene observed at high latitudes than other algorithms applied to AATSR measurements (Jafariserajehlou et. al., 2019).
We have added the description of the cloud identification algorithm of Jafariserajehlou et al. (2019) into section 2.3.1 "Cloud detection Algorithm (ASCIA)" of the revised manuscript. In particular, at line number 184 of the revised manuscript we have added "The ASCIA cloud detection algorithm achieved promising agreement of more than 95% and 83% within ±2 and ±1 okta when compared with ground-based synoptic surface observations (SYNOP) (WMO, 1995) over the Arctic.".

**Q3:** Many of the details in section 2.1.2 are given in a way that it is not possible to follow. For example: "As a result, PCC values calculated between multiple pairs of data for ground scenes of the same area at different times provide an indication of whether the scene is cloud-covered or cloud-free". Based on "multiple pairs of data" it is impossible to say if the correlation was computed for AOD, surface reflectance, top-of-atmosphere reflectance or something else.
**Response:**
The PCC values calculate reflectance at the top of the atmosphere between multiple pairs of data for ground scenes of the same area at different times give an indication of whether the scene is cloud-covered or cloud-free.
In section 2.3.1 "Cloud detection Algorithm (ASCIA)" of the revised manuscript at line number 144 we added "Consequently, the P CC values are derived using sets of reflectances at TOA of the same area at different times, which give an indication of whether the scene is cloudy or cloud-free (Lyapustin et al., 2008)".

**Q4:** I also asked for the cost function to be minimized in the retrieval to be shown in the manuscript. In the revised manuscript, the authors write: "The actual retrieval is working on Equations 7 and 4, and is minimized with respect to the optical depth (tau) at 555nm as a cost function." I do not understand what is meant by "minimizing equations 7 and 4" and would not be able reproduce the algorithm based on the details given in the manuscript.
**Response:**
In the newly created section 2.3.2 "Aerosol retrieval Algorithm" of the revised manuscript we now presented the cost function as Eq.18.

**Q5:** "The reflectance thresholds for a 3.7µm channel were used in the second part of ASCIA." However, no values for the thresholding are given and I cannot say if the authors mean top-of-atmosphere reflectance or surface reflectance.
**Response:**
We have added a brief description of the cloud identification algorithm of Jafariserajehlou et al. (2019) in section 2.3.1 "Cloud detection Algorithm (ASCIA)" in the revised manuscript. At line number 165 to 170 of the revised manuscript we have added the thresholds for a 3.7µm channel used in the second part of ASCIA for the top of the atmospheric (TOA) reflectance.

**Q6:** Overall, the section 2.1.2 is very difficult to follow and understand. I feel that based on the details given in the manuscript, it is not possible to replicate the results of the manuscript. It may be possible that everything was carried out correctly and following good scientific practices but I can not verify them based on this documentation shown in the manuscript.
**Response:**
We are sorry if the section 2.1.2 was difficult to follow. We have made changes in the revised manuscript to improve readability according to our explanations to Q1.

**Q7:** Furthermore, all the changes the authors have listed as proposed changes in the reply to referee have not been done in the actual manuscript as mentioned in the reply. For example: "At the end of line 19, we propose to add: The Arctic has experienced a significant increase in near-surface air temperatures over the past three decades: the rate of temperature increase being about four times larger than the global mean (Rantanen et al., 2022). This phenomenon is known as Arctic Amplification (AA)." This change has not been done in the revised manuscript.
**Response:**
We are sorry for this. We have corrected the mistake in the revised manuscript. We carefully rechecked the revised manuscript and have not found any missing entries any more.
**At line 22, we propose to modify:**
The Arctic has experienced a significant increase in near-surface air temperatures over the past three decades: the rate of temperature increase being about four times larger than the global mean (Rantanen et al., 2022). This phenomenon is known as Arctic Amplification (AA).

**Q8:** The authors claim both in the original and revised manuscript: "Uncertainties due to both space and time sampling differences are minimized". To be precise, just by using a certain collocation protocol (25km + 30min) does not minimize the uncertainties due to space and time. Also, minimize uncertainties of what?
**Response:**
We would like to remove the sentence "Uncertainties due to both space and time sampling differences are minimized" from our revised manuscript.
**At line 278, we propose to remove:**
Uncertainties due to both space and time sampling differences are minimized.

**Q9:** Finally, the authors justify some of their findings by citing an another manuscript under review in ACP. The ACP manuscript is written by the same authors and there they cite this AMT manuscript under review. I find it problematic to submit two quite similar manuscripts to two different journals simultaneously and cross-cite between the manuscripts many times and justify selections and findings in the other manuscript by the non-reviewed results shown in the other manuscript.

**Response:**

We would like to remove the citations of our ACP manuscript (https://doi.org/10.5194/egusphere-2023-730) from this AMT manuscript.

We would like to bring to the notice of the referee that, our ACP manuscript (https://doi.org/10.5194/egusphere-2023-730) is very different from the AMT manuscript.

In our AMT manuscript we presented the method as well as validation for the retrieval of the AOD for the pan-Arctic cryospheric region.

Whereas, in our ACP manuscript, we used GEOS-Chem global 3D chemical transport model simulations to find out that whether the aerosol components originating from natural and anthropogenic sources are attributed to AEROSNOW retrieved aerosol distributions over pan-Arctic sea ice region with high spatio-temporal coverage.

In addition, we have also performed a large-scale analysis focusing on the more spatio-temporal overage with regional, seasonal and annual variability of AOD components originating from anthropogenic as well as natural sources. Further, we investigated the impact of smoke intrusion events due to biomass burning, and seasonal change in precipitation on the aerosol variability and more importantly aerosol composition over the central Arctic cryospheric region.

**At line 302, we propose to remove the citation:**

This difference can be explained by using a chemical transport model by separating the total AOD to aerosol components (Breider et al., 2014)

**At line 344, we propose to remove the citation:**

The high anthropogenic aerosol loading (Arctic haze events) due to long-range transport (Willis et al., 2018) over Arctic snow and ice is captured by the AOD determined by AEROSNOW.

**At line 352, we propose to remove the citation:**

The promising AOD results obtained with AEROSNOW indicate that these can be used to evaluate and improve aerosol predictions for various chemical transport models (Willis et al., 2018), especially over the Arctic sea ice in spring and summer for the important period 2003-2011, which is within the period of Arctic amplification.

---

## Author Response (AR3)

**Referee #1**

**Title: Retrieval of aerosol optical depth over the Arctic cryosphere during spring and summer using satellite observations**

The authors would like to thank the reviewer for her/his interest and efforts to review our manuscript.

We hope that we have been able to answer satisfactorily the questions raised and clarify parts of the manuscript which were unclear or ambiguous.

In the following the referee comments and criticisms, our responses, as authors, and our resultant changes to the manuscript are colored black, blue and red respectively.

**Q1:** The revised version of the paper significantly enhances the clarity of the fundamental structure of the research. It is now evident that the primary achievement of this study lies in the fusion of two previously unrelated algorithms into AEROSNOW, while incorporating quality control measures. This combination allows the algorithm's application for the reliable retrieval of aerosol optical depth (AOD) over the Arctic region. Once the outstanding issues are addressed, this paper appears suitable for publication in the Atmospheric Measurement Techniques (AMT) journal.

Nevertheless, certain areas still require clarification. In the prior review, several questions arose due to the reviewers' misunderstanding of the content. Although the author guided the reviewers to relevant sections within the article, it highlights a fundamental issue with the paper - its difficulty in comprehension. While all the necessary information may be present, it lacks a logical organization that would enable someone unfamiliar with the research to grasp and follow it seamlessly.

As a proposed improvement, restructuring the paper is recommended. Section 2 should exclusively focus on data, encompassing descriptions of all data used in the algorithm, such as the MODIS data employed in the quality assurance (QA) process. Section 3, dedicated to the algorithm, should begin with a clear presentation of the algorithm's flowchart, emphasizing that this paper's core contribution is the development of a robust aerosol retrieval algorithm over the Arctic. This is achieved by amalgamating two existing algorithms and implementing dependable QA procedures, expanding the algorithm's applicability across a wider region and time span. Sections 3.1 and 3.2 can address the pre-existing algorithms, while Section 3.3 should expound upon the novel elements (notably, the "new contribution" seems to occur in the post-processing phase).

**Response:** According to the referee's suggestions, we have reorganized the Section 2 and 3 in the revised manuscript.

We have added "Space-borne observation: MODIS cloud products" in Sec.2.2 of the revised manuscript.

**Q2:** In the introduction, it is imperative to elucidate the limitations of exclusively using Istomina, 2009 and underscore the key advantages of uniting these two algorithms. Distinguishing between a research algorithm tailored to specific cases and a robust algorithm capable of operating across an entire region and data records is pivotal. The paper should emphasize this crucial distinction to effectively persuade its readers.

**Response:** We have presented the limitations of exclusively using Istomina et al., 2009, and advantages of uniting these two algorithms in the manuscript at line 58-63 and 72-74.

**We propose to modify the paragraph at line number 58 to 63 as follows to further highlight the advantages expected from uniting the two algorithms:** Several dedicated algorithms for passive satellite remote sensing over snow and ice have been developed. Istomina et al., 2009 and later Mei et al., 2013, 2020a, b have provided valuable pioneering research. However, these attempts have been mostly confined to the island of Spitsbergen in the Svalbard archipelago in northern Norway. Thus far, there have been no attempts to apply these algorithms together with the Arctic-adopted cloud masking algorithm systematically in the Arctic cryosphere to address the data gap identified above. Studies using active satellite remote sensing such as Sand et al. (2017) and Xian et al. (2021) are valuable, but the observational data are limited over the Arctic cryosphere.

**We propose to modify at line number 72 to 74 as follows:** We retrieve the total AOD using an approach first described by Istomina et al. (2009), which we have further integrated with the cloud masking algorithm of Jafariserajehlou et al. (2019) and named AEROSNOW. This AEROSNOW approach was then systematically applied over vast Arctic cryospheric regions.

**Q3:** I have some algorithm related questions as well, particularly concerning Istomina, 2009. Although this isn't the primary focus of your research, it's crucial due to its centrality in the algorithm. I'm not clear about where $\tau$ fits into the equations, which is the goal parameter. In my experience, $\tau$ should be in the $\rho_{atm}$ term in Eq. 3, which includes contributions from aerosol and Rayleigh and gas. Once you know how much is $\rho$ atm you can get $\tau$ using an assumed aerosol model.

**Response:** We agree with the referee and have added '$\tau$' in Eq.3 of the revised manuscript.

**Q4:** However, between line 249 to line 267, you describe two different methods of estimate $\rho_{atm}$. Many questions regarding these two paragraphs. First, is this $\rho$ atm the same in Eq. 4 vs. Eq. 3. If so, what is the point of having BRDF estimation if you already got $\rho_{atm}$. I assume this roughly estimated $\rho$ atm is only for atmosphere correction to get a better BRDF ratio, which is used in iterative process to fine tune the $\rho$ atm contribution. Eventually the $\rho$ atm in Eq. 3 will be the same as $\rho$ atm in Eq. 4. Not sure if my understanding is correct. Because it is not clearly stated in the paper. Second, both methods of estimating $\rho$ atm has their own uncertainties, one assuming coastal aerosol properties are the same with inland, another uses pre-defined aerosol models, plus assuming negligible ocean surface contribution, which also raises big concerns. It is not clear whether any of the method considered sedimentation or other watercolor contribution.

**Response:** We are sorry for the confusion. The aerosol retrieval algorithm presented in section 3.1.2 of the revised manuscript has undergone a comprehensive rewrite. Eq. 4 is derived from Eq. 3 and

$\rho_{atm}(\lambda, \mu 0, \mu, \varphi, \tau)$, which includes $\tau$,  is defined in Eq. 3, as the contribution of atmospheric reflectance to the one at the top of the atmosphere in the revised manuscript.

Further, two possible methods can be used for the estimation of $\rho_{atm}(\lambda, \mu 0, \mu, \varphi, \tau)$ but in Istomina et al., 2009, they have used Lookup table (LUT) approach for the retrieval.

We propose to remove the paragraph from line 249-256.

Other clarification questions:

**1.** It is still not clear to me how can ρ atm converted to τ without an aerosol model (or maybe it is the same model stated in Table 1 and Figure 2?).

**Response:** The response to this question was described in part in our answer to Q4.  In Istomina et al., (2009), SCIATRAN radiative transfer model has been used to calculate the look-up table.

This has been clarified at line number 270-274 of the aerosol retrieval algorithm section (Section 3.1.2) as follows, "The aerosol properties used in model simulations, such as the single scattering albedo (SSA ($\lambda$)), the real part, and the imaginary part of the refractive index for the coarse and accumulation modes of the water-soluble, oceanic, dust, and soot aerosol components are given in Table. 1 adopted from Istomina et al. (2011). Subsequently, a look-up table was calculated using the SCIATRAN radiative transfer model (Rozanov et al., 2014; Mei et al., 2023a). This LUT has been used for the determination of $\rho_{atm}(\lambda, \mu 0, \mu, \varphi, \tau )$".

**2.** The airmass factor and Angstrom equation discussion from line 269 to line 275 does not provide sufficient connection to the previous text.

As, we have rewritten the aerosol retrieval algorithm section (Section 3.1.2), we propose to remove the line from 269 to 275.

**3.** Eq. 18 is still very hard to understand because there is no specific definition of what ρ is, on the second term in the right hand side. If ρsfc is calculated from Eq. 15, then I assume ρ is from observation. But based on which equation, is it Eq. 3? Which the ρ in Eq. 4 is the ρ in Eq. 5? I assume is the left-hand side ρ. But when I read until Eq.15, I realized the ρ in Eq. 5 maybe ρ sfc,sim. To enhance clarity, it would be helpful to move Eq. 15 to the beginning and explain how you solve Eq. 15 instead of Eq. 4.

**Response:** In response to these questions, we have rewritten the aerosol retrieval algorithm section and hope that your questions are now answered.

To make it more clear we have removed the Eq.15 and formulated it more straight forward in Eq. 11 and Eq. 12 in the revised manuscript. As mentioned in question Q4, we have rewritten the aerosol retrieval algorithm section and hope that it is clearer and addresses your questions.

**5.** Line 241 to 249 discuss uncertainties in Eq. 4, which again, doesn't align with your approach completely, so it requires further discussion.

**Response:**  The response to this question was described in question Q4 in this authors response.

**Other points:**

**1.** The abstract should explicitly state that AEROSNOW is a result of merging two existing algorithms.

**We propose to add at line 7 to 8 in the abstract as follows:** AEROSNOW incorporates an existing aerosol retrieval algorithm with a cloud masking algorithm, alongside a novel quality flagging methodology specifically designed for implementation in the high Arctic region ( ≥ 72°N).

**2.** The paragraph that commences with "Recently, Toth et al., (2018)" around line 65 lacks a clear connection with the preceding content. This connection should be established earlier in the paragraph.

**We propose to modify the paragraph at line 64-68 as follows:** With respect to using active satellite remote sensing, recently, Toth et al. (2018) and Xian et al. (2021) reported that the active satellite sensor, the Cloud-Aerosol Lidar with Orthogonal Polarization (CALIOP/CALIPSO) (Winker et al., 2004) has a significant fraction of aerosol profile data comprising retrieval fill values (-9999s, or RFVs) and thus rejected. This is partly due to the minimal detection limits of the lidar when measuring the signal scattered back to space. In fact, in some areas of the Arctic, over 80 % of CALIOP profiles consist entirely of RFVs (Toth et al., 2018 and Xian et al., 2021).

**3.** The repetitive mention of "use AATSR to retrieve aerosols over the Arctic" should be minimized throughout the paper.

**Response:** This should not be the case anymore in the revised version of the manuscript.

**4.** Lines 93 and 95 both refer to "cloud masking." It's important to clarify that these are distinct cloud masking procedures, and a detailed explanation is necessary. However, if the suggested structural changes are implemented, this may no longer be an issue.

**Response:** We have made the structural changes in the revised manuscript as suggested by the referee in question Q2 of this author's response.

**5.** Ensure proper citation is provided in the AATSR data section.

**We propose to add the citation at line 99:** The AEROSNOW algorithm is applied to the dual view Level 1B data product reflectance at the top of the atmosphere made by AATSR (Llewellyn-Jones and Remedios, 2012).

**6.** The paragraph discussing the lack of other field campaign validation data should be relocated from the AERONET data section to the introduction within the Data section.

**We propose to move the paragraph to line 103-106:** In addition, the use of validation sources other than AERONET would have been very helpful. Unfortunately, data from valuable campaigns and expeditions such as POLAR-AOD (Mazzola et al., 2012), MOSAiC Expedition and MOSAiC-ACA (Mech et al., 2022), and AFLUX/PASCAL - Arctic (Mech et al., 2022) were only available after 2011.

**7.** In line 246, "but The ..." the capitalization of "T" in "The" is unnecessary.

**Response:** We are sorry for the typographical error, which has been rectified in the revised version.

**8.** Clarify whether this LUT is used for retrieval or solely for atmospheric correction in line 286.

**Response:** We hopefully clarified the use of LUT in our answer to question Q4.

**We propose to add at line 286:** This LUT has been used for the determination of $\rho_{atm}(\lambda, \mu0, \mu, \varphi, \tau)$.

**9.** Enhance the clarity of the flow chart by using color-coding to differentiate existing algorithms from new elements. The algorithm section should also encompass any additional pre-processing and post-processing steps applied, such as solar zenith angle (sza), snow-only filtering, and other quality control (QF) requirements.

**Response:** We have made changes in the flow chart as per the referee's suggestions in the revised manuscript.

**10.** Despite previous inquiries, there remains confusion regarding whether this algorithm is designed for partial snow cover or exclusively for 100% snow cover. Line 340 implies the consideration of snow cover fraction, suggesting applicability beyond 100% snow cover. However, line 225 states that "only pure snow-covered areas (100% snow cover) are used." This discrepancy requires clarification.

**Response:** At line number 340 the snow cover fraction is a variable name in MODIS Terra and Aqua data product, which is 100% snow cover. The AEROSNOW is used for AOD retrievals for scenes having 100% snow cover.

**11.** Specify whether the seasonal mean is calculated from level 2 data or monthly mean data.

**Response:** The seasonal mean is calculated from level 2 data.

**We propose to add at line number 365:** The seasonal mean is calculated by using level 2 data

**12.** Figure 7 still lacks standard deviation information; consider including it for greater completeness.

**Response:** We have added standard deviation information in Fig.7 as per the referee's suggestions.